# LARGE LANGUAGE MODEL ALIGNMENT VIA INVERSE REINFORCEMENT LEARNING FROM DEMONSTRATIONS

## ABSTRACT

Aligning Large Language Models (LLMs) is crucial for enhancing their safety and utility. However, existing methods, primarily based on preference datasets, face challenges such as noisy labels, high annotation costs, and privacy concerns. In this work, we introduce *Alignment from Demonstrations* (AfD), a novel approach leveraging high-quality demonstration data to overcome these challenges. We formalize AfD within a sequential decision-making framework, highlighting its unique challenge of missing reward signals. Drawing insights from forward and inverse reinforcement learning, we introduce divergence minimization objectives for AfD. Analytically, we elucidate the mass-covering and mode-seeking behaviors of various approaches, explaining when and why certain methods are superior. Practically, we propose a computationally efficient algorithm that extrapolates over a tailored reward model for AfD. We validate our key insights through experiments on the Harmless and Helpful tasks, demonstrating their strong empirical performance while maintaining simplicity.

## 1 INTRODUCTION

The alignment of Large Language Models (LLMs) is essential for their safe and effective deployment in various applications. Current research has focused extensively on reinforcement learning from human feedback (RLHF) (Christiano et al., 2017; Ouyang et al., 2022). However, the majority of advancements in RLHF (Rafailov et al., 2024b; Zhao et al., 2023; Yuan et al., 2023; Dong et al., 2023; Azar et al., 2023; Munos et al., 2023) rely on preference-based datasets annotated by humans or general-purpose LLMs (Bai et al., 2022; Lee et al., 2023; Guo et al., 2024), facing several significant challenges that can impede their performance or limit their applications:

1. **Noisy Labels Harm Alignment Performance**: Research indicates that noisier data leads to less accurate reward modeling and poorer alignment performance (Zheng et al., 2023). Since the same language model generates the response pairs in preference-based learning, the preferences provided by annotators can be highly uncertain and noisy (Azar et al., 2023).
2. **High Cost in Preference Annotation**: Although it is theoretically and empirically justified that the ideal approach to learning from preference data involves continuous querying of annotators during the learning process (Guo et al., 2024; Xiong et al., 2023; Touvron et al., 2023; Tang et al., 2024), this approach can be prohibitively expensive.
3. **Requirement of Inductive Biases in Reward Modeling**: Utilizing preference-based data often requires assumptions like the Bradley-Terry model (Bradley & Terry, 1952) or the Kahneman Tversky model (Ethayarajh et al., 2024). These assumptions may not always hold true, as discussed in (Azar et al., 2023; Munos et al., 2023; Swamy et al., 2024).
4. **Privacy Concerns in Preference Generation**: Collecting preference over data with the help of annotators or commercial general-purpose LLMs is not always feasible, particularly when dealing with private data that cannot be shared externally (Li et al., 2023; Pouplin et al., 2024).

To address these challenges, we propose aligning LLMs using a demonstration dataset, referred to as *Alignment from Demonstrations* (AfD), as an alternative to preference-based alignment. Specifically, AfD offers the following advantages: **(1)** demonstration data always enjoys higher quality and less noise; **(2)** AfD does not require continuous querying and comparison; **(3)** AfD does not rely on assumptions inherent in preference-based methods; **(4)** AfD enables LLM alignment without the need for external annotators, hence can be applied to private dataset locally.

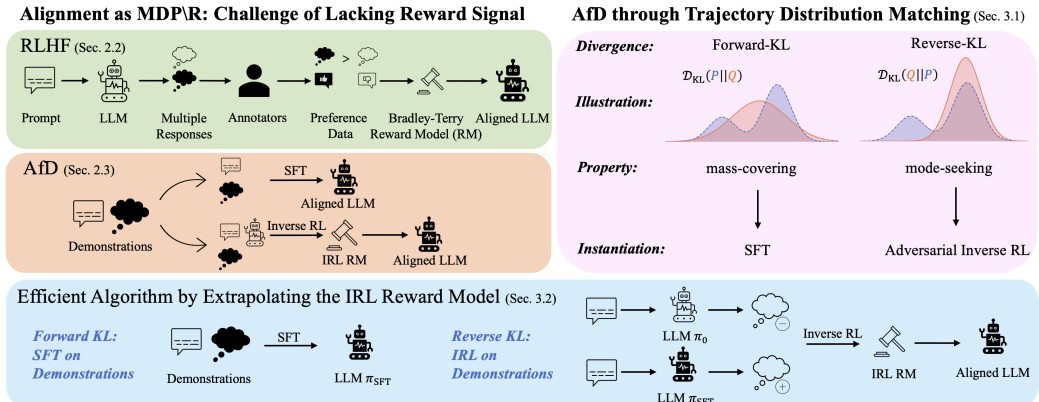

Figure 1: *A roadmap of this paper and comparison of different alignment approaches.* To address the challenges in preference-based alignment (Sec.1), we propose aligning LLMs using demonstration data. We first define the alignment problem as an MDP and disclose its challenge of lacking reward signals in Sec.2.2. In addition to the RLHF solution, we present alternative approaches from the perspective of RL (Sec.2.3). We then explore the trajectory distribution matching objective for AfD, connecting divergence measures with different algorithms (Sec.3.1). We introduce an efficient Inverse RL algorithm for the AfD problem in Sec.3.2. Experiments in Sec.4 empirically verify the proposed method and key insights. Related work is discussed in Appendix A.

Moreover, demonstration data is readily available in many real-world applications of LLMs. For instance, in medical AI systems, demonstrations might include desired diagnostics or prescriptions based on patients' electronic health records. In customer service chatbot systems, demonstrations could consist of dialogues between expert customer support agents and customers.

Despite the availability of such data, its use in aligning LLMs has typically been limited to supervised fine-tuning (SFT). In this work, we demonstrate that SFT corresponds to the Behavior Cloning method that applies demonstration datasets in reinforcement learning (RL). Moreover, we systematically explore the potential of demonstration datasets from a formal RL perspective, providing both theoretical rationales and empirical evidence on how to exploit these datasets for aligning LLMs.

**To highlight the main contributions and take-aways of our work:**

1. Conceptually, we demonstrate the superiority of AfD, which addresses the challenges inherent in conventional preference-based alignment. We formally define the AfD problem using a sequential decision-making framework and connect it with previous practices in Inverse RL to enhance understanding of potential solutions.
2. Methodologically, we introduce the trajectory distribution matching objectives for AfD. Within this unified objective framework, we show that SFT and adversarial learning are both trajectory-matching utilizing different divergences. This sheds light on the mass-covering and mode-seeking behaviors attainable through various divergences.
3. Practically, we identify the challenge of reward hacking in AfD, explaining why naively applying reward modeling may fail in the context of alignment. We propose an easy-to-implement algorithm to address this issue effectively.
4. Empirically, we validate our proposed insights and methods on the `Harmless` and `Helpful` splits of the `Anthropic HH-RLHF` dataset (Bai et al., 2022) through comprehensive comparisons with existing methods and ablation studies.

## 2    ALIGNMENT BEYOND PREFERENCE DATA AND SUPERVISED FINE TUNING

In this section, we present our central insight: the LLM alignment problem can be framed within the context of *forward and inverse* RL, suggesting it can be addressed using corresponding methodologies. To ensure this section is self-contained, we provide the necessary preliminaries and background concepts in the gray text boxes . The section is organized as follows: In Section 2.1, we elaborate on the sequential decision-making nature of auto-regressive LLM generation. In Section 2.2, we discuss the challenge of missing reward signals in LLM alignment and the difficulties associated with current solutions. In Section 2.3, we present the perspective that AfD can be formulated as an Inverse RL problem, highlighting the potential solutions from such a perspective.

## 2.1 Auto-Regressive Language Generation as Sequential Decision Making

We first cast auto-regressive language generation into the Markov Decision Processes framework for sequential decision-making.

> **Markov Decision Processes (MDP)** In Markov Decision Processes, decisions are made in discrete time steps and affect the state of the environment in the subsequent step. Formally, an MDP is denoted as $\mathcal{M} = \{\mathcal{S}, \mathcal{A}, \mathcal{T}, \mathcal{R}, \rho_0, \gamma\}$, where $\mathcal{S} \subset \mathbb{R}^d$ denotes the $d$-dim state space, $\mathcal{A}$ is the action space. Broadly, the environment includes $\mathcal{T}$ and $\mathcal{R}$, the former denotes the transition dynamics $\mathcal{T} : \mathcal{S} \times \mathcal{A} \mapsto \Delta(\mathcal{S})$ that controls transitions between states, and the reward function $\mathcal{R} : \mathcal{S} \times \mathcal{A} \mapsto \mathbb{R}$ provides feedback. $\rho_0 = p(s_0) \in \Delta(\mathcal{S})$ denotes the initial state distribution. $\gamma$ is the discount factor that trades off between short-term and long-term returns.

In the context of the token-generation process in LLMs, let $C$ denote the context window size and $\mathcal{V}$ denote the vocabulary, including the special tokens like [EOS] and [MASK]. The MDP is instantiated as follows: State space $\mathcal{S} = \mathcal{V}^C$; action space $\mathcal{A} = \mathcal{V}$; transition dynamics is **deterministic and known**: $s' = \mathcal{T}(s,a) = \texttt{Concat}(s,a) = [s,a]$; We consider states containing an [EOS] token as absorbing states, meaning $\forall a : s' = \mathcal{T}(s,a) = s$ if [EOS] $\in s$; an LLM $\ell$, serving as policy $\pi = \ell$, generates the next token $a \in \mathcal{A}$ based on the current context $s \in \mathcal{S}$; The initial state distribution of queries is $\rho_0$, and $T$ represents the maximal number of new tokens in a generation. i.e., $T$ is the maximal number of transitions in the MDP. For instance, in the following case, the context window length $C \geq 7$ and $T = 2$, an initial state $s_0$ is given as follows:

$$s_0 = \big[\ \texttt{The}\ |\ \texttt{color}\ |\ \texttt{of}\ |\ \texttt{the}\ |\ \texttt{sky}\ |\texttt{[MASK]}|\texttt{[MASK]}\big],$$

when the language model policy $\pi$ selects a new token "is" from the vocabulary $\mathcal{V}$, the next state deterministically becomes

$$s_1 = \texttt{Concate}(s_0, a_0 = \texttt{is}) = \big[\ \texttt{The}\ |\ \texttt{color}\ |\ \texttt{of}\ |\ \texttt{the}\ |\ \texttt{sky}\ |\ \texttt{is}\ |\texttt{[MASK]}\big],$$

the generation process continues until either the [EOS] token is selected, the maximal context window size is reached, or the maximal decision steps $T$ is reached. In this example, the final generated context could be:

$$s_2 = \texttt{Concate}(s_1, a_1 = \texttt{blue}) = \big[\ \texttt{The}\ |\ \texttt{color}\ |\ \texttt{of}\ |\ \texttt{the}\ |\ \texttt{sky}\ |\ \texttt{is}\ |\ \texttt{blue}\ \big].$$

## 2.2 Challenge of the Alignment MDP: Getting Reward Signals is Hard

The research on LLM alignment focuses on aligning language models with users' intentions during response generation (Ouyang et al., 2022). Within the MDP framework, users' intentions are represented by a reward model $\mathcal{R}$, which provides feedback on the LLM's outputs, evaluating aspects such as helpfulness, truthfulness, and harmlessness of the generated content. Typically, evaluations are performed at the trajectory level, meaning feedback is provided only after the entire generation process is complete:

$$\mathcal{R}(s_t, a_t) = \begin{cases} r(s_t) & \text{if } s_t \text{ is a terminal state, } t = T \\ 0 & \text{otherwise.} \end{cases} \tag{1}$$

Ideally, human users would provide feedback for each response, allowing conventional online RL algorithms to optimize the policy $\pi = \ell$ through

$$\pi^* = \arg\max_{\pi \in \Pi} \mathbb{E}_{a_t \sim \pi, s_{t+1} \sim \mathcal{T}, s_0 \sim \rho_0} \sum_{t=0}^{T} \gamma^t \mathcal{R}(s_t, a_t) = \arg\max_{\pi \in \Pi} \mathbb{E}_{a_t \sim \pi, s_{t+1} \sim \mathcal{T}, s_0 \sim \rho_0} r(s_T), \tag{2}$$

However, a significant challenge in LLM alignment is **the difficulty in defining reward signals**, as the desired user intentions are not easily accessible. In prevailing LLM alignment approaches, reward models are typically derived from preference-based annotations.

**Learning Reward Models from Preference Annotations.** Most recent advancements in LLM alignment rely on preference-based datasets of the form $\mathcal{D}_{\text{pref}} = \{x_i, y_i^+, y_i^-\}_{i \in [N]}$, where $y_i^+$ and $y_i^-$ are the preferred and dis-preferred responses given input $x_i$. Models such as Bradley-Terry (Bradley & Terry, 1952) are then used to convert ranking feedback into absolute scores to serve as reward signals. Thus, we call the reward model built with a preference-based dataset the Bradley-Terry Reward Model (BT-RM). As has been discussed earlier, these datasets pose several challenges,

including noisy labels (Azar et al., 2023; Zheng et al., 2023), high costs (Guo et al., 2024; Xiong et al., 2023; Touvron et al., 2023; Tang et al., 2024), the requirement of additional assumptions in transferring rank to scores (Azar et al., 2023; Munos et al., 2023; Bradley & Terry, 1952; Ethayarajh et al., 2024; Rafailov et al., 2023) [1], and privacy concerns.

## 2.3 DEMONSTRATIONS AS AN ALTERNATIVE TO PREFERENCE-BASED DATA FOR ALIGNMENT

In RL research, learning from human feedback through preference is not the only option when reward signals are unknown or difficult to design (Plappert et al., 2018). Learning from a demonstrative behavioral dataset has been widely applied in various domains, including robotics control (Schaal, 1996; Nair et al., 2018; Hester et al., 2018), autonomous driving (Kuderer et al., 2015; Scheel et al., 2022), video game playing (Vinyals et al., 2019), and AlphaGo (Silver et al., 2016). Formally, with a demonstration dataset containing paired states and high-quality actions: $\mathcal{D}_{\text{demo}} = \{s_i, a_i^*\}_{i \in [N]}$, the most direct approach, Behavior Cloning (Pomerleau, 1991), learns the policy through supervised learning:

> **Behavior Cloning (BC)** A demonstrative decision dataset is collected from a behavior policy $\pi_\beta$. Denoting the state-action pairs in the dataset as $(s_i, a_i^*)$, the BC method learns a policy through a supervised learning objective:
> $$\pi_{\text{BC}} = \arg\max_\pi \mathbb{E}_{(s_i, a_i) \sim \mathcal{D}_{\text{demo}}} \log(\pi(a_i|s_i))$$

**Supervised Fine Tuning: Behavior Cloning for AfD.** In the context of LLM alignment, demonstrations in the form of $\mathcal{D}_{\text{SFT}} = \{x_i, y_i^*\}_{i \in [N]}$ are also referred to as the Supervised Fine Tuning (SFT) dataset. This format is versatile: for example, $x$ can be a general query for Question-Answering tasks, an incomplete sentence for completion tasks, or a general instruction for instruction following tasks; Correspondingly, $y^*$ represents the desired answers, a completed sentence, or a response following the instruction. Such datasets are widely applied for SFT training, where the learning objective is to minimize the token-wise difference given the existing context. To clarify our notations for further discussion, consider the following example of a context-response pair $x_i, y_i^*$:

$$x_i = \left[\begin{array}{c|c|c|c|c|c|c} \text{What} & \text{is} & \text{the} & \text{color} & \text{of} & \text{the} & \text{sky?} \end{array}\right],$$
$$y_i^* = \left[\begin{array}{c|c|c|c|c|c|c} \text{The} & \text{color} & \text{of} & \text{the} & \text{sky} & \text{is} & \text{blue} \end{array}\right].$$

the SFT training first reorganizes the dataset $\mathcal{D}_{\text{SFT}}$ to state-action pairs ($\mathcal{D}_{\text{demo}}$) as follows:

$$s_0 = \left[\begin{array}{c|c|c|c|c|c|c|c|c|c|c} \text{What} & \text{is} & \text{the} & \text{color} & \text{of} & \text{the} & \text{sky?} & \text{[MASK]} & \text{[MASK]} & \text{[MASK]} & \text{...} \end{array}\right],$$
$$a_0^* = \boxed{\text{The}},$$
$$s_1 = \left[\begin{array}{c|c|c|c|c|c|c|c|c|c} \text{What} & \text{is} & \text{the} & \text{color} & \text{of} & \text{the} & \text{sky?} & \text{The} & \text{[MASK]} & \text{[MASK]} & \text{...} \end{array}\right],$$
$$a_1^* = \boxed{\text{color}},$$
$$s_2 = \left[\begin{array}{c|c|c|c|c|c|c|c|c|c} \text{What} & \text{is} & \text{the} & \text{color} & \text{of} & \text{the} & \text{sky?} & \text{The} & \text{color} & \text{[MASK]} & \text{...} \end{array}\right],$$
$$a_2^* = \boxed{\text{of}},$$
$$...$$

with such a dataset, the learning objective is to reproduce the demonstration token $a_i^*$ when the LLM (policy) is given $s_i$ (incomplete token sequences). The training of the SFT is conducted through supervised classification.

**AfD Beyond Supervised Fine Tuning.** While BC is conceptually simple and easy to implement, it faces a fundamental challenge known as the *distributional shift* — during evaluation, the state distribution is generated by rolling out the learned policy $\pi$, rather than the data-generation behavior policy $\pi_\beta$. To address this challenge, Imitation Learning (IL) and Inverse RL consider scenarios where the *dynamics model* is available to generate roll-out samples during learning (Pomerleau, 1991; Finn et al., 2016; Abbeel & Ng, 2004). For a more detailed discussion on the benefits of accessing dynamics models, refer to Appendix C.1.

At first glance, aligning LLMs with an offline demonstration dataset might seem like an offline RL problem, as no further interactions with human annotators are available during training. However, it is the accessibility of online interactions with the ***dynamics model***, rather than the reward model, that

---

[1] see further analysis in Appendix B

determines the online or offline nature of the tasks. In LLM alignment practices, while accessing reward models (online annotators) during training is impossible, **the dynamics model in response generation is known and accessible** — the actions are tokens generated by LLMs, and the responses (trajectories) are concatenations of those generated tokens. This insight naturally leads us to explore alternative approaches rooted in the IL and Inverse RL literature. In Table 3 of Appendix A.4, we contextualize the difference and link between various topics in the RL literature.

Building on the notations and connections established above, we now introduce a unified objective class using trajectory distribution matching, a widely studied objective in the IL and Inverse RL literature (Jarrett et al., 2020; Ho & Ermon, 2016; Ghasemipour et al., 2020), for the AfD problem.

# 3 ALGORITHMS FOR ALIGNMENT FROM DEMONSTRATIONS

## 3.1 ALIGNMENT FROM DEMONSTRATION THROUGH TRAJECTORY DISTRIBUTION MATCHING

Unlike the action distribution matching objective used in BC, when the dynamics model is accessible, it is beneficial to study the occupancy matching problem to enhance the performance of learning from the offline demonstrations (Ho & Ermon, 2016; Ross et al., 2011; Fu et al., 2017; Orsini et al., 2021). Specifically, we denote the state-action occupancy measure of the behavior policy (i.e., the demonstrator) as $\rho^\beta(s,a) = \pi_\beta(a|s) \sum_{t=0} \gamma^t \text{Prob}(s_t = s|\pi_\beta)$, and the state-action occupancy measure of the current policy as $\rho^\pi(s,a)$. Intuitively, the occupancy measure describes the distribution of state-action pairs visited by an agent under a given policy. For auto-regressive LLMs that take context $x$ as input and output response $y = (y^{(0)}, y^{(1)}, ..., y^{(T)} = \text{EOS})$ containing a maximum of $T+1$ tokens, we have

$$\begin{aligned}
\rho^\pi(s_k, a_k) &= \rho^\pi(s_k = (x, y^{(0:k-1)}), a_k = y^{(k)}) \\
&= \pi(a_k = y^{(k)}|s_k = (x, y^{(0:k-1)}))p(s_k) \\
&= \pi(a_k = y^{(k)}|s_k = (x, y^{(0:k-1)}))\pi(a_{k-1} = y^{(k-1)}|s_{k-1} = (x, y^{(0:k-2)}))p(s_{k-1}) \quad (3) \\
&= ... \\
&= p(s_0)\Pi_{t=0}^{t=k}\pi(a_t = y^{(t)}|s_t = (x, y^{(0:t-1)}))
\end{aligned}$$

In alignment, we are motivated to study the completed generations. Therefore, it is useful to denote the trajectory distribution $d^\pi(y|x)$ as the occupancy measure of completed generations conditioned on input context $x$ (i.e., final state occupancy conditioned on initial state):

$$d^\pi(y|x) = \Pi_{t=0}^{t=T}\pi(a_t = y^{(t)}|s_t = (x, y^{(0:t-1)})) = \rho^\pi(s_T, a_T)/p(x) \quad (4)$$

Practically, we can sample from the above conditional distribution by rolling out the policy $\pi$, and approximately sample from the behavior policy using the demonstration dataset:

$$d^\beta(y|x) = \Pi_{t=0}^{t=T}\pi_\beta(a_t = y^{(t)}|s_t = (x, y^{(0:t-1)})) = \rho^\beta(s_T, a_T)/p(x) \quad (5)$$

In the following, we derive different objectives for LLM alignment from the perspective of divergence minimization between the demonstration conditional distribution and the roll-out conditional distribution. Specifically, we study the minimization of Forward KL-Divergence and Reverse KL-Divergence in the main text, as they are the most commonly used and provide sufficient insights into the proposed objectives. We additionally discuss a more general framework in Appendix D.

**AfD through Divergence Minimization using Forward KL.** We first consider the objective using the forward KL divergence between the demonstration and policy conditional trajectory distributions:

$$\min_\pi \left[ \text{KL}(d^\beta(y|x)||d^\pi(y|x)) \right] = -\max_\pi \mathbb{E}_{(x,y)\sim\mathcal{D}_{\text{SFT}}} \left[ \log d^\pi(y|x) \right]$$

$$= -\max_\pi \mathbb{E}_{(x,y^{(0:K)})\sim\mathcal{D}_{\text{SFT}}} \left[ \sum_{t=0}^K \log \pi(a_t|s_t) \right]. \quad (6)$$

Comparing the derived objective with the SFT objective, which minimizes the negative log-likelihood of tokens in the demonstration dataset given the existing context:

$$\min_\pi \mathbb{E}_{(s,a)\sim\rho^\beta} \left[ \text{KL}(\pi^\beta(a|s)||\pi(a|s)) \right] = -\max_\pi \mathbb{E}_{(s,a)\sim\mathcal{D}_{\text{demo}}} \left[ \log(\pi(a|s)) \right] \quad (7)$$

we find that both approaches yield exactly the same learning objective.

> **Take-Aways:** Using the forward KL in **conditional trajectory distribution divergence minimization** leads to the same objective as SFT, where the training objective minimizes the KL divergence of **action marginal distribution** between $\pi^\beta$ and $\pi$. The forward KL divergence is known to result in mass-covering behavior, whereas the reverse KL divergence leads to mode-seeking behavior (Ghasemipour et al., 2020; Khalifa et al., 2020; Wiher et al., 2022; Wang et al., 2023; Gu et al., 2023). This equivalence explains the mass-covering behavior observed in SFT in recent literature Kirk et al. (2023).

**AfD through Divergence Minimization using Reverse KL.** In the pursuit of mode-seeking behavior, we can minimize the Reverse KL divergence, leading to the following learning objective:

$$\min_\pi [\mathrm{KL}(d^\pi(y|x) || d^\beta(y|x))] = \min_\pi \mathbb{E}_{(x,y) \sim d^\pi} \left[ \log d^\pi(y|x) - \log d^\beta(y|x) \right]. \tag{8}$$

The challenge with this objective is that the second term, $d^\beta(y|x)$, is always unknown. This issue has been addressed in the literature through adversarial training (Fu et al., 2017). By training a discriminative model $D_\phi$, parameterized by $\phi$, to classify trajectories sampled from the demonstration dataset or the behavior policy $\pi$, we achieve

$$D_\phi^*(y|x) = \frac{d^\beta(y|x)}{d^\beta(y|x) + d^\pi(y|x)} \tag{9}$$

at optimal convergence (Goodfellow et al., 2014). Plugging Equation (9) into Equation (8) (a proof can be found at Appendix D.1), we derive a practical policy learning objective:

$$\max_\pi \mathbb{E}_{(y|x) \sim d^\pi} \left[ \log D_\phi(y|x) - \log(1 - D_\phi(y|x)) \right] \tag{10}$$

The discriminative mode $D_\phi$ can be optimized through:

$$\max_\phi \mathbb{E}_{(y|x) \sim \mathcal{D}_{\mathrm{SFT}}} [\log D_\phi(y|x)] + \mathbb{E}_{(y|x) \sim d^\pi} [\log(1 - D_\phi(y|x))] \tag{11}$$

> **Take-Aways:** Comparing the learning objectives derived using the reverse KL divergence to the SFT objective, we see that performing mode-seeking is generally more challenging than mass-covering due to the **difficulty of estimating the probability of trajectory from the demonstrator**. This challenge can be circumvented through adversarial training.

Despite its success, adversarial training is known to be unstable and computationally expensive (Salimans et al., 2016; Kodali et al., 2017; Lin et al., 2021; Yang et al., 2022a), which is particularly concerning when applied to training LLMs in the AfD context. In the next section, we leverage insights from the adversarial objective discussed above to propose a computationally efficient algorithm that avoids iterative training.

### 3.2 Efficient Inverse RL by Extrapolating Over Reward Models

Conceptually, the optimization of policy in Equation (10) is conducted by maximizing over the inner variable, sharing the same form as Equation (2). This observation suggests using the reward notation:

$$r(y|x) = \log D_\phi(y|x) - \log(1 - D_\phi(y|x)) \tag{12}$$

Specifically, when $D_\phi(y|x)$ is instantiated by neural networks with sigmoid activation function over logits $D_\phi(y|x) = \sigma(\texttt{logits}(y|x))$, we have $r(y|x) = \texttt{logits}(y|x)$ — the reward signal is provided by the discriminative model through its output logits. In the following discussion, we interchangeably use the terms reward model and discriminative model as they refer to the same concept. We call this reward model the Inverse-RL Reward Model, abbreviated as IRL-RM.

Inspired by the previous success achieved in the Inverse RL literature that extrapolates learned reward models (Brown et al., 2019), we propose to circumvent the difficulty in iterative generative adversarial training through reward model extrapolation. Initially, one might build a reward model using samples from the demonstration dataset as positive examples and samples generated by the initial LLM policy as negative examples for discriminator training.

Table 1: *Comparison of multiple reward modeling choices.* The first three rows are choices in building reward models in AfD using different datasets for the discriminative model training.

| Dataset for RM | Negative Example Source | Positive Example Source | Format of Data | Heterogeneity in RM |
|---|---|---|---|---|
| Init-SFT RM | $(y|x) \sim \pi_{\text{init}}$ | $(y|x) \sim \pi_{\text{SFT}}$ | AfD | Low |
| Init-Demo RM | $(y|x) \sim \pi_{\text{init}}$ | $(y|x) \sim \mathcal{D}_{\text{demo}}$ | AfD | High |
| SFT-Demo RM | $(y|x) \sim \pi_{\text{SFT}}$ | $(y|x) \sim \mathcal{D}_{\text{demo}}$ | AfD | High(er) |
| Preference-based | Dispreferred | Preferred | Pair-wise | No |

Nevertheless, in the AfD problem, the demonstration dataset is typically generated by external demonstrators, such as human experts or more advanced LLMs, rather than the LLM being aligned. This **heterogeneity** can introduce significant bias in the reward modeling step, potentially leading to reward hacking (Skalse et al., 2022; Gao et al., 2023; Zhang et al., 2024; Coste et al., 2023). The reward model may focus on the heterogeneity of responses — for discrimination — rather than on the informative aspects that truly evaluate the quality of responses in terms of human intention.

It is important to note that in our context, the reward model is trained to differentiate the origins of various responses. **A discriminator that primarily detects subtle differences due to model heterogeneity is not effective as a reward model for providing meaningful improvement signals for alignment.**

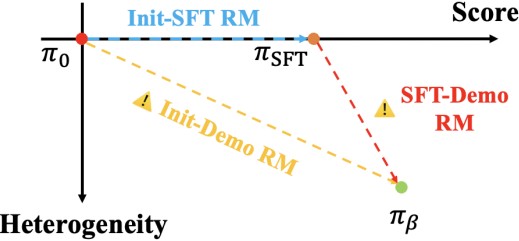

To address this challenge, we propose using a different dataset format for building our reward model. Instead of using the demonstration dataset as positive samples, we use the samples generated by the SFT policy $\pi_{\text{SFT}}$, trained on the demonstration dataset, as positive examples. The samples generated by the initial LLM policy $\pi_0$ serve as negative examples. This approach alleviates the heterogeneity issue that arises when naively combining demonstration samples with $\pi_0$-generated samples. Table 1 contrasts the different data choices for reward model training. Figure 2 visualizes and illustrates their differences. To further explain and contrast different approaches:

Figure 2: *Illustration of different choices for positive and negative samples in Inverse-RL reward modeling.* The LLM to be aligned is restricted to a specific model class, limiting its expressivity and capability. This limitation is depicted by allowing improvements only along the x-axis. For example, SFT training on the demonstration dataset can push the initial model $\pi_0$ toward higher scores. The y-axis represents the heterogeneous nature of the demonstration dataset in AfD problems, where the behavior policy $\pi_\beta$ always differs from the LLM to be aligned. Notably, $\pi_\beta$ could be human experts or stronger general-purpose LLMs.

- **Init-Demo RM**: Using samples generated by $\pi_0$ as negative examples and demonstration dataset samples as positive examples in reward model training is straightforward. However, as $\pi_0$ and $\pi_\beta$ are heterogeneous models, so nuanced differences, such as specific verb usage or response formats in $\pi_\beta$ can dominate reward model learning rather than the desired alignment properties.
- **SFT-Demo RM**: Using samples generated by $\pi_{\text{SFT}}$ examples and demonstration dataset samples as positive examples faces the same challenge. Moreover, since $\pi_{\text{SFT}}$ and $\pi_\beta$ are closer in terms of the desired properties to align (scores), reward hacking is even more likely.
- **Init-SFT RM**: To avoid potential reward hacking caused by using heterogeneous data in reward model training, we can use samples generated by $\pi_0$ as negative examples and samples generated by $\pi_{\text{SFT}}$ as positive examples. Unlike the previous approaches, where positive and negative examples are generated by heterogeneous models, these two models are homogeneous since the SFT policy is fine-tuned from the initial policy.
- **Preference-based RM** (BT-RM): In preference-based reward modeling, both preferred and dispreferred responses are samples from the same LLM (Ouyang et al., 2022). Therefore, there is no issue of heterogeneity between the positive and negative samples.

When applying the learned reward models at inference time to determine which responses are superior, these responses are generated by $\pi_{\text{SFT}}$, therefore, the **Init-SFT RM** should outperform other choices. We provide pseudocode in Appendix E.1. As has been insightfully pointed out by our reviewer, we discuss an alternative closed-form reward model in Appendix G for interested readers. Before using empirical studies to verify our findings, it is worth further discussing the related work on the superiority of having explicit reward models at the end of this section.

### 3.3 EXTENDED DISCUSSIONS ON DPO AND SPIN

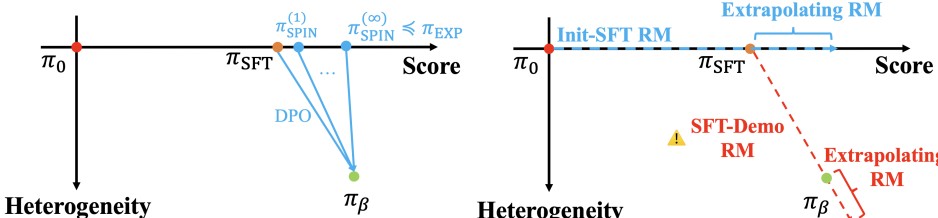

Figure 3: Illustration on the difference between SPIN and extrapolation over learned RMs in Inverse RL.

It is worth noting the links and differences between our explicit reward modeling approach and the direct alignment methods such as DPO (Rafailov et al., 2023) that are designed for alignment using demonstration datasets (Chen et al., 2024).

Regardless of the data format, DPO-type algorithms have a crucial distinction: they explicitly assume the existence of a score-based scalar reward derived from the Bradley-Terry model, which requires pair-wise data for effective application. In contrast, adversarial learning approaches utilizing discriminative models do not rely on such explicit assumptions about the Bradley-Terry model or the preference-based data format. While constraining the reward function to a specific form may mitigate the reward ambiguity issue in inverse RL (Fu et al., 2017; Ng et al., 2000; 1999; Chan et al., 2024), it also limits the expressivity of the reward space. Adversarial imitation approaches, as introduced in our work, do not presuppose any specific reward model form. This allows for a broader range of alternatives to the Bradley-Terry model, including direct preference objectives (Azar et al., 2023; Munos et al., 2023) and prospect theory objectives (Ethayarajh et al., 2024).

Moreover, DPO does not extrapolate over its implicit reward model. When applying DPO iteratively to the demonstration dataset, as proposed in Chen et al. (2024), the underlying assumption is that the current policy (starting with the SFT policy) is always weaker than the demonstrations. Hence, the Bradley-Terry model can be repeatedly applied to these pairwise data. At the convergence of iterative training (Chen et al., 2024), the performance of the aligned LLM is upper-bounded by the performance of the demonstrations, as the demonstration dataset is consistently regarded as the positive examples in implicit reward modeling. The first panel of Figure 3 illustrates the learning objectives and how policies evolve during learning.

Our method, as an Inverse RL approach, explicitly learns the reward model and extrapolates over it. As illustrated in the right panel of Figure 3, our reward modeling mechanism extrapolates the reward model based on task scores. Conversely, if we naively follow the SPIN setup — using the SFT checkpoint's generation as negative examples and the demonstrations as positive examples — the generated reward model can negatively impact heterogeneity, leading to poor performance. We will empirically demonstrate this point in our experimental section.

This distinction highlights the flexibility and potential of our approach in **achieving super-demonstration performance** in LLM alignment, as will be empirically verified in the next section.

## 4 EXPERIMENTS

**Overview.** In this section, we validate the insights and methods proposed in earlier discussions. Our experiments are designed to: (1) Demonstrate the efficacy of alignment from demonstrations and verify the insights derived from the Inverse RL perspective (Sec. 4.1). (2) Evaluate the necessity and performance of the proposed reward modeling method (Sec.4.2). (3) Assess the scalability and effectiveness of the reward model in policy optimization, highlighting the feasibility of alignment without preference-based data (Sec.4.3). Due to space limit, we defer more comprehensive of results including PPO as policy optimizers in Appendix E.7

**Tasks.** To evaluate the performance of our proposed methods, we focus on the `Harmless` and `Helpful` tasks from the Anthropic HH-RLHF dataset Bai et al. (2022). Demonstrations were generated using the OpenAI GPT-4 API, with detailed prompting strategies available in Appendix

E.2. The Harmless task includes 42.5K training examples and 2.3K testing examples. Due to the content-filtering feature of the GPT-4 API, we got 25.6K responses as the demonstration dataset. For the Helpful task, which comprises 43.8K training examples and 2.3K testing examples, our demonstration dataset includes 42.7K examples gathered from the API.

**Base Models and Evaluation Metrics.** For the Harmless task, we employ GPT-2 Radford et al. (2019) as our base model, given its potential in alignment and its capability of output harmless responses. For the Helpful task aimed at enhancing the helpfulness of responses, we utilize the more advanced Gemma model Team et al. (2024) at the 2B parameter scale, tailored to our hardware specifications. Our evaluation employs two metrics to measure the alignment efficacy of different methodologies: golden reward model scoring and GPT4-as-a-critic evaluation. In the golden reward model evaluation, we report on the reward scores as assessed by publicly available golden reward models Dong et al. (2023); Lambert et al. (2024); Yang et al. (2024). In the GPT4-as-a-critic evaluation, we use GPT-4 to evaluate which of the two responses more effectively meets the alignment criteria of a given query. More details can be found in Appendix E.

### 4.1 AfD with Forward KL: Supervised Fine Tuning

**Experiment Setup.** In this section, we aim to verify the effectiveness of aligning LLMs from demonstrations and the insight we draw from the Inverse RL perspective. We assess and compare the performance of the following single-phase training methods: **SFT-AfD**: Utilizes the demonstration dataset for supervised fine-tuning; **SFT-Preferred**: Employs supervised fine-tuning using the positive samples from the preference-based dataset; **DPO-Preference**: the Direct Preference Optimization method working on the preference-based annotations Rafailov et al. (2023); **DPO-AfD**: Represents a naive baseline that applies DPO directly to the demonstration dataset, treating samples generated by the initial policy as negative samples. Additionally, we benchmark the performance of the **Basemodels** prior to training and normalize the scores against the quality of the **Demonstrations**. All implementations are executed using the TRL library von Werra et al. (2020). To ensure fair comparisons, hyperparameters across different methods are standardized, with detailed configurations available in Appendix E.

**Results.** As depicted in Figure 4, the golden reward model evaluations for both tasks show promising results. In the Harmless task, SFT on the demonstration dataset not only matches but exceeds the performance of the demonstrator [2]. For both tasks, DPO on the demonstration dataset proves more effective than its application on the preference dataset. However, SFT applied only to the positive

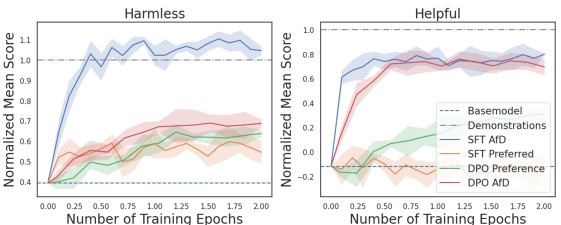

Figure 4: *Evaluation results using golden reward models.*

samples from the preference dataset shows negligible improvement in task performance. To keep the main text and the main takeaway message of this section clear, we deferred an extended version of Figure 4 to include full results comparing against all methods in Appendix E.7.

> **Take-Aways.** AfD proves to be a promising single-phase approach for alignment. In the Harmless task, where the response modes are limited, SFT demonstrates exceptional performance, affirming its equivalence to trajectory distribution matching using forward KL divergence. Nevertheless, SFT does not reach the same level of performance as the demonstrator in the Helpful task, where response variability is greater. Subsequent sections will explore the enhancement of AfD through reward modeling.

### 4.2 Building Effective Reward Models using Demonstrations

**Experiment Setup.** We now verify the effectiveness of the proposed RMs. We consider the four reward models discussed in Sec. 3.2: the **Init-SFT RM**; the **Init-Demo RM**; the **SFT-Demo RM** and the **Human-Pairwise** (the preference-based BT-RM) — as a reference. We use the Best-of-N (BoN) approach which stably archives on-par performance to the state-of-the-art policy optimization algorithms according to the literature Dong et al. (2023); Gao et al. (2023); Coste et al. (2023), maximally isolating and highlighting the sources of improvement.

---

[2]The demonstrator GPT4 rejects to answer (filters) some of the harmful queries on the test set.

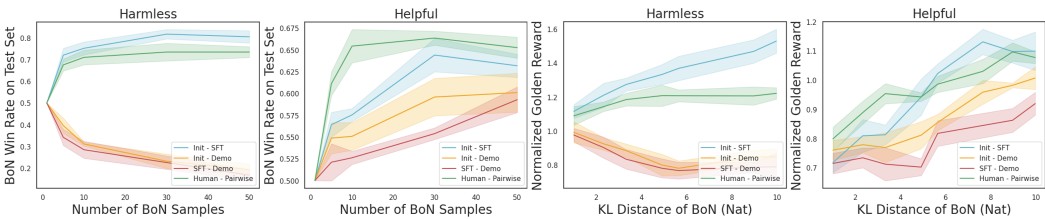

Figure 5: *Evaluating choices of building reward models using golden reward models.*

**Results.** For comparative analysis, we utilize the golden reward model. Specifically, the first two panels of Figure 5 illustrate the **Win Rates** of selected samples to be better than a deterministic generation. The latter two panels detail the normalized golden reward scores as the number of $N$ in BoN increases.

> **Take-Aways.** The results underscore the efficacy of building reward models using the demonstration dataset. Notably, the IRL RM using the **Init-SFT** stands out by achieving the highest win rates and scores compared to other models. Its performance matches or surpasses the preference-based reward model — yet the IRL RM can work without preference annotations.

### 4.3 BOOSTING PERFORMANCE BY EXTRAPOLATING REWARD MODELS

**Experiment Setup.** To further verify our method, we employ GPT4-as-a-judge as an additional metric. We stress-test the performance of the proposed reward models at large KL-divergence ($\approx$ 10 Nat) from the original SFT policy. We compare BoN using the proposed reward model (**BoN IRL-RM**), BoN using preference dataset (**BoN BT-RM**), and the **SFT** checkpoint.

Table 2: GPT4-as-a-critic evaluation on the BoN policies using different reward models and the SFT checkpoint.

| | Task | Harmless | | | Helpful | | |
| --- | --- | --- | --- | --- | --- | --- | --- |
| | | BoN IRL-RM | BoN BT-RM | SFT | BoN IRL-RM | BoN BT-RM | SFT |
| BoN IRL-RM | Win | - | 0.422(18) | 0.677(16) | - | 0.318(16) | 0.932(8) |
| | Tie | - | 0.351(17) | 0.147(12) | - | 0.298(15) | 0.039(6) |
| | Lose | - | 0.227(15) | 0.176(13) | - | 0.383(16) | 0.029(5) |
| BoN BT-RM | Win | 0.227(15) | - | 0.486(18) | 0.383(16) | - | 0.943(7) |
| | Tie | 0.351(17) | - | 0.260(16) | 0.298(15) | - | 0.036(6) |
| | Lose | 0.422(18) | - | 0.254(15) | 0.318(16) | - | 0.021(5) |
| SFT | Win | 0.176(13) | 0.254(15) | - | 0.029(5) | 0.021(5) | - |
| | Tie | 0.147(12) | 0.260(16) | - | 0.039(6) | 0.036(6) | - |
| | Lose | 0.677(16) | 0.486(18) | - | 0.932(8) | 0.943(7) | - |

**Results.** Table 2 presents the findings. The BoN strategy using the IRL RM markedly outperforms the SFT baseline. Notably, the performance of the IRL RM matches that of the preference-based RM, with the advantage of being developed solely from the demonstration dataset.

> **Take-Aways.** Employing the IRL RM in conjunction with the BoN strategy substantially enhances the performance of SFT policy in AfD. This improvement is particularly significant in the Helpful task, where the mass-covering property of SFT proves insufficient. These results are refreshing, demonstrating that AfD is a viable and effective alternative to RLHF.

## 5 CONCLUSION

In this paper, we addressed the limitations of preference-based alignment for Large Language Models (LLMs) by proposing an alternative approach: Alignment from Demonstrations (AfD). Our study highlights the benefits of using high-quality demonstration data, which avoids the noise, cost, and assumptions inherent in preference-based methods, and privacy concerns. By framing the AfD problem within a sequential decision-making framework and introducing trajectory distribution matching objectives, we provide a solid foundation for AfD. Our empirical results, validated on the `Harmless` and `Helpful` tasks of the `Anthropic HH-RLHF` dataset, demonstrate the effectiveness of AfD in achieving superior alignment performance. This work establishes AfD as a viable and efficient alternative to Reinforcement Learning from Human Feedback (RLHF), paving the way for safer and more reliable deployment of LLMs in various applications.

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

APPENDIX: TABLE OF CONTENTS

## A    RELATED WORK

### A.1    IMITATION LEARNING AND INVERSE REINFORCEMENT LEARNING

In contrast to the prevailing approaches in LLM alignment research, which rely on preference datasets, this work focuses on offline expert demonstration datasets. These datasets are more accessible in real-world applications and serve as the basis for developing algorithms that can surpass the performance of Supervised Fine-Tuning (SFT), the common practice for such datasets. The use of demonstration datasets, combined with the accessibility of the dynamics model, naturally frames the problem as an Imitation Learning (IL) or Inverse Reinforcement Learning (Inverse RL) task.

The simplest approach to IL in the literature is Behavior Cloning (BC) (Pomerleau, 1991), which leverages supervised learning to predict the actions in the demonstration dataset given the states. However, this method is often unreliable due to compounding errors (Ross et al., 2011). Adversarial Imitation Learning algorithms (Ho & Ermon, 2016; Fu et al., 2017; Ghasemipour et al., 2020; Kostrikov et al., 2018; Orsini et al., 2021), inspired by both Generative Adversarial Networks (GANs) (Goodfellow et al., 2014) and Inverse RL (Ng et al., 2000; Ziebart et al., 2008), aim to solve this problem by matching distributional characteristics. Specifically, GAIL seeks to learn a policy whose state-action space occupancy measure is indistinguishable from that of the expert demonstrations. A key difference between Inverse RL and IL is whether or not the reward model is explicitly modeled (Fu et al., 2017). With a learned reward model, the objective can go beyond matching demonstration behavior to extrapolating the reward model for super-demonstration performance (Brown et al., 2019).

There are two unique properties in the LLM alignment Markov Decision Process (MDP) that differentiate it from conventional IL and Inverse RL literature:

1. **Known and Deterministic Transition Dynamics:** In LLM alignment, the transition dynamics are known and deterministic, allowing us to explicitly define the trajectory distribution and use *trajectory distribution matching as the learning objective*.

2. **Sparse Reward Signals:** The reward signal is provided and is mostly meaningful only at the trajectory level, making the *alignment problem a sparse-reward IL task*. This sparsity means that learning a step-wise reward function, as done in existing work (Fu et al., 2017), may not be feasible.

### A.2    REINFORCEMENT LEARNING FROM HUMAN FEEDBACK

Introduced in the seminal paper by (Christiano et al., 2017), Reinforcement Learning from Human Feedback (RLHF) provides an alternative to traditional scalar reward signals in policy learning. In the context of LLMs, (Ouyang et al., 2022) proposed a three-step alignment framework consisting of SFT, reward modeling (RM), and policy learning with proximal policy optimization (PPO). This framework relies on two distinct types of datasets: 1. the SFT dataset contains queries and expert-generated responses to those queries, under the form of $\mathcal{D}_{\text{demo}} = \{x_i, y_i^*\}_{i \in [N_e]}$; and 2. the preference dataset $\mathcal{D}_{\text{pref}} = \{x_i, y_i^+, y_i^-\}_{i \in [N_p]}$ that contains queries, multiple language model responses, and human preferences over those response labeled by human annotators.

Current RLHF practices adhere to this two-stage, two-dataset framework, with several enhancements introduced in recent literature. For instance: the DPO circumvents explicit reward modeling and stabilizes the learning process on preference dataset using supervised signals (Rafailov et al., 2024b); SLiC-HF (Zhao et al., 2023) gains insight from contrastive learning and learns from closed-form losses that maximize the margin between the preferred and dispreferred generations; other alternatives include iterative supervised learning (Yuan et al., 2023; Dong et al., 2023), regularizing the generation (Azar et al., 2023) or game-theory-based methods (Munos et al., 2023; Chen et al., 2024; Cheng et al., 2023). These advancements collectively contribute to refining the RLHF framework, addressing various challenges associated with preference-based alignment in LLMs. Different from those approaches, our work focuses on *Alignment from Demonstrations*, where only a single demonstration dataset is used.

A.3 GENERATIVE ADVERSARIAL NETWORKS ON TEXT GENERATION MODELS

The use of GANs in text generation is also relevant to our research. Specifically, TextGAIL (Wu et al., 2021) explores GAN training for text generation that surpasses supervised learning performance. Other notable works using GANs for sequence generation include (Yu et al., 2017; Ke et al., 2019; Che et al., 2017; Guo et al., 2018; Adiwardana et al., 2020; Zhou et al., 2020; Caccia et al., 2018), all of which focus on text domain sequence generation.

Our work diverges from this line of literature in several key ways:

1. **Focus on Alignment**: Unlike GAN-based text generation, which often aims to generate context under specific formats (e.g., story generation), our work focuses on aligning LLMs to human intentions rather than merely generating text.

2. **Objective Comparison**: GAN-based methods are more akin to adversarial imitation techniques, aiming to reproduce the training dataset's distribution (Ho & Ermon, 2016). In contrast, our objective is to improve language model alignment by learning a reward model inspired by Inverse Reinforcement Learning (IRL) (Fu et al., 2017).

3. **Evaluation Metrics**: In many GAN-based text generation tasks (Wu et al., 2021; Yu et al., 2017; Ke et al., 2019; Che et al., 2017; Guo et al., 2018; Adiwardana et al., 2020; Zhou et al., 2020; Caccia et al., 2018), oracle evaluation metrics are available, eliminating the need to infer the underlying intention of demonstrations. In alignment tasks, however, human intention is not directly accessible as a function, necessitating a different approach.

4. **Motivation, Formulation, and Explanation**: Our work is motivated by the challenge of lacking reward signals in LLM alignment, formulated as an RL problem. We derive objectives from IRL literature to explain when and why SFT and IRL techniques are effective.

5. **Practical Implementation**: Unlike GAN-based methods, which rely on iterative training, our implementation does not. Instead, we extrapolate the learned IRL reward model (Brown et al., 2019) to further enhance the performance of SFT-ed LLMs.

This differentiation highlights our unique approach to LLM alignment, focusing on the nuances of reward modeling and alignment objectives, distinct from traditional GAN-based text generation methods.

A.4 COMPARISON OF DIFFERENT SET-UPS OF RL

In this section, we contextualize the differences and connections among various RL problem setups. Specifically, we discuss (online) RL, Offline-RL, Imitation Learning, Inverse-RL, Learning from Demonstrations, and Preference-based RL.

To elaborate on Table 3, we outline the following distinctions:

• Online RL: In this setup, both the external dynamics model and the reward model are accessible. An agent learns through trial and error by interacting with these live models.

• Offline RL: Neither the dynamics model nor the reward model is available. The agent learns solely from an offline dataset that includes information on states, actions, rewards, and transitions.

• Imitation Learning (IL): The reward model is unknown, but the dynamics model is accessible. The agent learns from demonstrations to optimize its policy, without explicitly modeling the reward.

• Inverse RL (IRL): Similar to IL, the reward model is unknown, but the dynamics model is accessible. The agent learns from demonstrations with the objective of building an explicit reward model to guide policy optimization.

• Offline IRL: Both the dynamics model and the reward model are unknown. The agent must learn from an offline dataset that contains demonstrations, but without direct access to the dynamics or reward models.

• Learning from Demonstrations: An agent initially uses an offline demonstration dataset to warm-start, followed by learning through online interactions with the environment, which includes the dynamics and reward models.

Table 3: Summarizing difference in problem settings of RL, Offline-RL, Imitation Learning (IL), Inverse-RL, Offline Inverse-RL (Offline IRL), Learning from Demonstrations (LfD), and Preference-based RL.

| Problem Settings | External Dynamics Model | External Reward Model | Learned Reward Model | Demo-nstration | Examples Solvers |
|---|---|---|---|---|---|
| RL | ✓ | ✓ | ✗ | ✗ | PPO (Schulman et al., 2017), TD3 (Fujimoto et al., 2018), SAC (Haarnoja et al., 2018) |
| Offline-RL | ✗ | ✗ | ✓ or ✗ | ✓ | BC (Pomerleau, 1991), CQL (Kumar et al., 2020), WGCSL (Yang et al., 2022b) |
| Imitation | ✓ | ✗ | ✗ | ✓ | BC (Pomerleau, 1991), AOC (Sun et al., 2023), GAIL (Ho & Ermon, 2016) |
| Inverse-RL | ✓ | ✗ | ✓ | ✓ | BC (Pomerleau, 1991), AIRL (Fu et al., 2017) |
| Offline-IRL | ✗ | ✗ | ✓ | ✓ | BC (Pomerleau, 1991), AOC (Sun et al., 2023), SBIL (Jarrett et al., 2020) |
| LfD | ✓ | ✓ | ✗ | ✓ | DQNfD (Hester et al., 2018), DDPGfD (Nair et al., 2018), AlphaStar (Vinyals et al., 2019) |
| Preference-based RL | ✓ | ✗ | ✓ | Paired | CPL (Hejna et al., 2023), T-REX (Brown et al., 2019), RLHF (Christiano et al., 2017; Ouyang et al., 2022), DPO (Rafailov et al., 2023) |

- Preference-based RL: This setup is similar to the inverse RL setting, but instead of a demonstration dataset, it uses a paired preference dataset. The Bradley-Terry model can transform ranking information into reward values, enabling the agent to learn from preferences rather than direct demonstrations.

Our method, which builds a reward model using the demonstration dataset, falls into the class of **Inverse RL** settings. By understanding these distinctions, we can better appreciate the nuances of each RL setup and their applicability to various problems in reinforcement learning.

## A.5  EXTENDED EMPIRICAL RESULTS ON SPIN

**Empirical Verification**  we experimented on the Harmless and Helpful tasks, with the results detailed in Table 4 below. Our IRL-RM method enhances performance beyond demonstration quality on both datasets, yet SPIN does not achieve this super-demonstrator performance.

Table 4: **Comparisons to SPIN**. IRL-RM achieves super-demonstration performance, yet SPIN does not.

| Task | Demo | SFT | IRL (N=10) | (N=30) | (N=50) | SPIN (iter=1) | (iter=2) | (iter=3) |
|---|---|---|---|---|---|---|---|---|
| Harmless | 1.704 | 1.785 | 2.171 | 2.272 | 2.333 | 1.769 | 1.753 | 1.747 |
| Helpful | 0.735 | 0.588 | 0.598 | 0.692 | 0.751 | 0.640 | 0.699 | 0.706 |

## A.6  COMPARING DIRECT AND EXPLICIT REWARD MODELING METHODS IN ALIGNMENT

In the following, we compare the direct alignment methods with explicit reward modeling methods from the perspective of generalization.

Explicit reward modeling has been shown to generalize to OOD samples better than direct alignment methods, both theoretically and empirically (Lin et al., 2024; Xu et al., 2024). The central insight behind this advantage lies in the observation that learning a discriminative model, such as a reward model that evaluates the quality of responses, can often be easier than learning a generative model that produces complete outputs (Ouyang et al., 2022). Such an efficiency arises from the fact discriminative models only need to capture the decision boundaries in identifying response qualities, rather than modeling the full response distribution.

# B ASSUMPTIONS BEHIND THE PREFERENCE-BASED (BRADLEY-TERRY) REWARD MODELING

The Bradley-Terry model (Bradley & Terry, 1952) and Elo score (Elo & Sloan, 1978) were originally developed for rating chess players, where the pairwise competition logs are switched to absolute scores.

**The Gaussian Assumption on Performance** To be specific, the Bradley-Terry model assumes the ability of **players** can be expressed as a score. In each two-player game, each player's performance will be a Gaussian distribution centered at this score. The variances of those Gaussian distributions are induced by the stochastic nature of the game and variability of the players' performance.

For instance, when player $A$ having score $S_A$ variance $\sigma_A$ and player $B$ having score $S_B$ variance $\sigma_B$ are playing against each other in a game, the probability that $A$ wins $B$ ($A \succ B$) in a game given the above Gaussian assumption on performance gives the following result:

$$P(A \succ B) = P\left(x_a \geq x_b | x_a \sim N(S_A, \sigma_A^2), x_b \sim N(S_B, \sigma_B^2)\right) = \frac{1}{2} + \frac{1}{2}\mathrm{erf}\left(\frac{S_A - S_B}{\sqrt{2(\sigma_A^2 + \sigma_B^2)}}\right) \tag{13}$$

In practice, other sigmoid-type functions besides the error function $\mathrm{erf}(\cdot)$ can be used, e.g., using $\tanh(\cdot)$ when assuming the distribution is logistic.

**Bradley-Terry Model in LLM Alignment** When it comes to RLHF, the Bradley-Terry model is applied to transfer **preference-based data** into scores. In such a process, the human evaluation is noisy, and the probability of observing response $y_A$ to be preferred over response $y_B$ is expressed as

$$P(y_A \succ y_B | x) = \frac{1}{2} + \frac{1}{2}\tanh\left(\frac{r_A - r_B}{\sqrt{2(v_A^2 + v_B^2)}}\right) \tag{14}$$

where $v_A, v_B$ models the variation in evaluating the value of different responses, and $r_A, r_B$ are the corresponding standardized scores of response $y_A, y_B$ given query $x$, respectively.

In principle, there are two functions to be estimated given a preference dataset $\mathcal{D}_{\mathrm{pref}} = \{x_i, y_i^+, y_i^-\}_{i \in [N]}$.

1. First, the reward function $R_\theta : \mathcal{X} \times \mathcal{Y} \mapsto \mathbb{R}$ evaluates how good an answer $y \in \mathcal{Y}$ is for a query $x \in \mathcal{X}$. e.g., $r_A = R_\theta(x, y_A), r_B = R_\theta(x, y_B)$.
2. Second, the variation function $V_\phi : \mathcal{X} \times \mathcal{Y} \mapsto \mathbb{R}$ evaluates how hard it is to evaluate whether an answer $y \in \mathcal{Y}$ is for a query $x \in \mathcal{X}$ is better than the other. e.g., $v_A = V_\phi(x, y_A), v_B = V_\phi(x, y_B)$.

Using the Cross-Entropy Loss to fit $\mathcal{D}_{\mathrm{pref}}$, we have

$$\mathcal{L}_{\mathrm{CE}} = -\mathbb{E}_{(x,y^+,y^-) \sim \mathcal{D}_{\mathrm{pref}}}\left[\log \sigma\left(\frac{R_\theta(x, y^+) - R_\theta(x, y^-)}{\sqrt{(V_\phi^2(x, y^+) + V_\phi^2(x, y^-))/2}}\right)\right] \tag{15}$$

In the common practice of RLHF based on the Bradley-Terry model (Christiano et al., 2017; Ouyang et al., 2022; Rafailov et al., 2024b), the learning of reward model only focuses on the score and eliminates the variation in evaluation. Therefore, the denominator is **simplified by setting** $V_\phi^2(x, y^+) = V_\phi^2(x, y^-) = 1$, i.e., the score is normalized by the variation of the problem.

$$\widetilde{\mathcal{L}}_{\mathrm{CE}} = -\mathbb{E}_{(x,y^+,y^-) \sim \mathcal{D}_{\mathrm{pref}}}\left[\log \sigma\left(R_\theta(x, y^+) - R_\theta(x, y^-)\right)\right] \tag{16}$$

The Bradley-Terry model in RLHF assumes human annotators' preference can be expressed as scores centered at the real scores of different responses, yet it differs from the Bradley-Terry model used in chess rating or games in the sense that

1. The RLHF dataset contains queries from different domains, some of which are intrinsically harder to evaluate, hence directly using the B-T model is to some extent like using a unified rating system of chess, Go, and poker — the scores are not well calibrated.

2. Different from chess, where the `number of players` $\ll$ `number of games`, in RLHF, the number of players (query-response pairs) is comparable to the number of games (annotator comparison).

3. The Elo scores are executed and updated in an online manner, and offline learning with preference-based data may lose the ability to error correction. Among those challenges, (1) and (2) can potentially be addressed with a learned variance term in the B-T model.

# C  EXTENDED PRELIMINARIES

## C.1  ONLINE AND OFFLINE RL

**Online RL**  In the *Online RL* setting, an agent with policy $\pi \in \Pi : \mathcal{S} \mapsto \Delta(\mathcal{A})$ learns through trial and error. It actively interacts with the environments — including both transition dynamics $\mathcal{T}$ and the reward function $\mathcal{R}$.

At each time step $t$, an agent observes a state $s_t$ from the environment and selects an action $a_t \sim \pi$. Upon taking the action, the agent receives a reward $r_t$ and transit to a new state $s_{t+1}$. The agent's objective is to maximize its expected return.

$$\pi^* = \arg\max_{\pi \in \Pi} \mathbb{E}_{a_t \sim \pi, s_{t+1} \sim \mathcal{T}, s_0 \sim \rho_0} \sum_{t=0}^{T} \gamma^t \mathcal{R}(s_t, a_t), \tag{17}$$

We can alternatively denote the trajectory generated by a policy $\pi$ to be $\tau = \{s_0, a_0 \sim \pi(a_0|s_0), s_1 \sim \mathcal{T}(s_1|s_0, a_0), a_1 \sim \pi(a_1|s_1), ...\}$ and denote the trajectory distribution of $\pi$ as

$$p_\pi(\tau) = \rho_0 \Pi_{t=0}^{T} \pi(a_t|s_t) \mathcal{T}(s_{t+1}|s_t, a_t), \tag{18}$$

where $T$ denotes the length of decision sequences. The learning objective can be expressed as

$$\pi^* = \arg\max_{\pi} \mathbb{E}_{\tau \sim p_\pi(\tau)} \left[ \sum_{t=0}^{T} \gamma^t \mathcal{R}(s_t, a_t) \right]. \tag{19}$$

**Offline RL**  In the *Offline RL* setting, interactions with the environment are strictly forbidden. The learning problem is no longer online learning but learning from a static dataset of decision logs $\mathcal{D}_{\text{Offline}} = \{(s_t^i, a_t^i, s_{t+1}^i, r_t^i)\}$, that is generated by some unknown behavior policy $\pi_\beta$.

The most obvious difficulty in the offline RL setting is such a setting prohibits exploration — hence it hinders the improvement of policy learning to be improved over the demonstration data.

## C.2  BEHAVIOR CLONE AND IMITATION LEARNING

**Behavior Cloning (BC)**  Assuming the decision dataset is collected from an optimal behavior policy $\pi_\beta^*$, every decision $a_t^i$ is optimal. Denoting the state-action pairs in the dataset as $(s_t, a_t^*)$, the BC method learns a policy through a supervised learning objective that minimizes the difference between decision demonstration pairs. i.e.,

$$\pi = \arg\min_{\pi} \mathbb{E}_{(s_t^i, a_t^i) \sim \mathcal{D}} ||a_t^i - \pi(s_t^i)||^2 \tag{20}$$

A fundamental challenge of BC is the *distributional shift*: in evaluation, the state distribution is sampled from rolling out the learned policy $\pi$, rather than the behavior policy $\pi_\beta$ that generates the dataset.

then the expected number of mistakes made by the learned policy $\pi$ based on such an expert decision dataset can be denoted as

$$\ell(\pi) = \mathbb{E}_{p_\pi(\tau)} \left[ \sum_{t=0}^{T} \mathbb{1}(\pi(s_t) \neq a_t^*) \right] \tag{21}$$

Then we have the following theorems:

**Theorem C.1** (Behavior Clone Error Bound. (Ross et al., 2011))**.** *If $\pi$ is trained via empirical risk minimization on $s_t \sim p_{\pi_\beta}(\tau)$ and optimal labels $a_t^*$, and attains generalization error $\epsilon$ on $s_t \sim p_{\pi_\beta}(\tau)$, then $\ell(\pi) \leq C + T^2 \epsilon$ is the best possible bound on the expected error of the learned policy.*

*Remark* C.2 (Compounding Error.). An intuitive interpretation of this quadratic relationship between the error bound and the generalization error is that those errors aggregate along the trajectory. i.e., whenever the learned policy makes a mistake, it tends to make more mistakes from then on as that action is not optimal and will lead to other out-of-distribution states, which will lead to further mistakes.

*Remark* C.3 (Behavior Clone). We can always set up a supervised learning objective in offline RL to minimize the difference between decision demonstration pairs. i.e.,

$$\pi = \arg\min_\pi \mathbb{E}_{(s_t^i, a_t^i) \sim \mathcal{D}} ||a_t^i - \pi(s_t^i)||^2 \tag{22}$$

**Imitation Learning (IL)**   In order to alleviate the challenge of compounding error we discussed above, IL considers the setting where a dynamics model is available during learning. The objective of IL is to learn from a (decision) demonstration dataset, with access to a dynamics model — such that the **current policy can be rolled out in the real environment**. Intuitively, with such a dynamics model, the optimization objective will no longer be $s_t \sim p_{\pi_\beta}(\tau)$ but could be $s_t \sim p_\pi(\tau)$ — **the distributional shift problem can be alleviated.** It has been shown in the literature that having access to a *dynamics model* is essential in controlling the error bound. (Ross et al., 2011)

There are many practical methods for implementing such a learning process, and the most famous work in the Deep-RL era is the GAIL (Ho & Ermon, 2016), which conducts IL through adversarial learning: the policy is a *generator* of behaviors, while a *discriminator* then tries to identify whether a trajectory is generated by the behavior policy $\pi_\beta$ or by the generator (the policy learned).

**Theorem C.4** (DAgger Error Bound, (Ross et al., 2011)). *If $\pi$ is trained via empirical risk minimization on $s_t \sim p_\pi(\tau)$ and optimal labels $a_t^*$, and attains generalization error $\epsilon$ on $s_t \sim p_\pi(\tau)$, then $\ell(\pi) \leq C + T\epsilon$ is the best possible bound on the expected error of the learned policy.*

*Remark* C.5. This requires the additional assumption of being able to access the behavior (expert) policy $\pi_\beta$ actively to acquire the expert for those roll-out trajectories generated by $\pi$ .

# D    GENERAL DISTRIBUTIONAL MATCHING FRAMEWORK USING $f$-DIVERGENCE

Formally, according to the $f$-divergence framework of GANs (Nowozin et al., 2016) and Inverse RL (Ghasemipour et al., 2020), the alignment problem can be written as training an LLM model $\pi$, such that

$$\min_\pi \max_{T_\omega} \mathbb{E}_{(s,a) \sim \mathcal{D}_{\text{demo}}}[T_\omega(s,a)] - \mathbb{E}_{(s,a) \sim \pi}[f^*(T_\omega(s,a))] \tag{23}$$

where $f : \mathbb{R}^+ \mapsto \mathbb{R}$ is a convex, lower-semicontinuous function, and it defines a statistical divergence between distribution $P, Q$ with density function $p, q$ as: $D_f(P||Q) = \int_x q(x) f\left(\frac{p(x)}{q(x)}\right) dx$, and $f^*$ is the conjugate of $f$, defined as $f^* = \sup_{u \in \text{dom}_f}\{ut - f(u)\}$. Practically, it was shown in (Ghasemipour et al., 2020) that Equation (23) can be solved through iterative optimizing

$$\max_{T_\omega} \mathbb{E}_{(s,a) \sim \mathcal{D}_{\text{demo}}}[T_\omega(s,a)] - \mathbb{E}_{(s,a) \sim \pi}[f^*(T_\omega(s,a))] \tag{24}$$

and

$$\max_\pi \mathbb{E}_{\tau \sim \pi}[\sum_t f^*(T_\omega(s_t, a_t))] \tag{25}$$

To elaborate on how different choices of $f$ lead to different practical implementations of the AIL approach of alignment, we take the state-action occupancy measure here for example:

- AIRL: $f(u) = -\log(u)$ ;                   $D_f(\rho^{\text{demo}}||\rho^\pi) = \text{KL}(\rho^\pi||\rho^{\text{demo}})$
- GAIL: $f(u) = -(u+1)\log\frac{1+u}{2} + u\log u$;   $D_f(\rho^{\text{demo}}||\rho^\pi) = \text{JS}(\rho^\pi||\rho^{\text{demo}})$
- FAIRL: $f(u) = u\log(u)$;                 $D_f(\rho^{\text{demo}}||\rho^\pi) = \text{KL}(\rho^{\text{demo}}||\rho^\pi)$
- $\alpha$-IRL: $f(u) = \frac{u^{1-\alpha} - (1-\alpha)u - a}{\alpha(\alpha-1)}$;           $D_f(\rho^{\text{demo}}||\rho^\pi) = D_\alpha(\rho^{\text{demo}}||\rho^\pi)$

Therefore, the methods discussed in the main text could be extended to other divergences in the $f$-Divergence framework. Moreover, the discussion in the main text focused on trajectory distribution matching. Another potential learning objective is state-action distribution matching. We provide the following results, yet those objectives assume token-level feedback (Chan et al., 2024; Rafailov et al., 2024a). We leave the investigation of their empirical performance to future work.

### D.1 PROOF OF EQUATION(9)

For any LLM generator $\ell$, which takes input $x$ and output a response $y$, the training criterion is to maximize the value function $V(\ell, D_\phi)$ defined as

$$
\begin{aligned}
V(\ell, D_\phi) &= \int_{(x \oplus y)} p_\beta(x \oplus y) \log(D_\phi(x \oplus y)) d(x \oplus y) + \int_x p_\beta(x) \log(1 - D_\phi(\ell(x))) dx, \\
&= \int_{(x \oplus y)} [p_\beta(x \oplus y) \log(D_\phi(x \oplus y)) + p_\pi(x \oplus y) \log(1 - D_\phi(x \oplus y))] d(x \oplus y),
\end{aligned}
\tag{26}
$$

For any $(a, b) \in \mathbb{R}^2 \backslash \{0, 0\}$, the function $u \mapsto a \log(u) + b \log(1 - u)$ achieves its maximum in $[0, 1]$ at $\frac{a}{a+b}$, with the disciminator defined on the joint support of $p_\beta$ and $p_\pi$. Conditioning the generation on the training prompt of the dataset, we get Equation 9. □

### D.2 ALIGNMENT WITH THE STATE-ACTION MATCHING USING THE FORWARD KL-DIVERGENCE

From the perspective of **trajectory distribution matching**, the Forward-KL minimization directly leads to the BC objective — maximizing the likelihood of a trajectory under the learner's policy is equivalent to maximizing the likelihood of each action taken in that trajectory.

On the other hand, when using the **occupancy measure** instead of the trajectory in distribution matching, connecting Forward-KL with BC needs additional information on the dynamics model. In the context of token-generation MDPs, this leads to a **weighted version** of the BC objective.

When minimizing the forward KL divergence between **state-action occupancy measures**

$$
\min_\pi \left[ \mathrm{KL}(\rho^{\mathrm{demo}}(s, a) || \rho^\pi(s, a)) \right] = - \max_\pi \mathbb{E}_{(s,a) \sim \rho^{\mathrm{demo}}} \left[ \log \rho^\pi(s, a) \right] \tag{27}
$$

$$
= - \max_\pi \mathbb{E}_{(s_k, a_k) \sim \rho^{\mathrm{demo}}} \left[ \log \Pi_{t=0}^k \pi(a_t | s_t) \right] \tag{28}
$$

$$
= - \max_\pi \mathbb{E}_{(s_k, a_k) \sim \rho^{\mathrm{demo}}} \left[ \sum_{t=0}^k \log \pi(a_t | s_t) \right] \tag{29}
$$

$$
= - \max_\pi \mathbb{E}_{(s_k, a_k) \sim \rho^{\mathrm{demo}}} \left[ \frac{T - k}{T} \log \pi(a_k | s_k) \right] \tag{30}
$$

Minimizing the forward KL divergence of **state-action occupancy measure** is different from the SFT objective by a re-weighting factor, depending on the **position of the token in the demonstration sequence**. Intuitively, it can be understood as a re-weighting approach to avoid compounding errors.

### D.3 ALIGNMENT WITH THE STATE-ACTION MATCHING USING THE REVERSE KL-DIVERGENCE

When considering the reverse KL divergence on the **state-action occupancy measure**, the learning objective is

$$
\min_\pi [\mathrm{KL}(\rho^\pi(s, a) || \rho^{\mathrm{demo}}(s, a))] = - \max_\pi \mathbb{E}_{(s,a) \sim \rho^\pi} \left[ \log \rho^\pi(s, a) - \log \rho^{\mathrm{demo}}(s, a) \right]. \tag{31}
$$

The difficulty in the above learning objective is that the second term is always unknown. In the literature, such a difficulty has been solved through adversarial training (Fu et al., 2017). By training a discriminative model $D_\phi$ parameterized by $\phi$ that learns to classify state-actions sampled from the demonstration dataset or from the behavior policy $\pi$, we get

$$
D_\phi^*(s, a) = \frac{\rho^{\mathrm{demo}}(s, a)}{\rho^{\mathrm{demo}}(s, a) + \rho^\pi(s, a)} \tag{32}
$$

at its optimal convergence (Goodfellow et al., 2014). Plugging Equation (32) into Equation (31), an practical policy learning objective can be given by

$$\min_{\pi} \mathbb{E}_{(s,a)\sim\rho^{\pi}} \left[\log D_{\phi}(s,a) - \log(1 - D_{\phi}(s,a))\right] \tag{33}$$

and $D_{\phi}$ is optimized iteratively through:

$$\max_{\phi} \mathbb{E}_{(s,a)\sim\rho^{\text{demo}}}[\log D_{\phi}(s,a)] + \mathbb{E}_{(s,a)\sim\rho^{\pi}}[\log(1 - D_{\phi}(s,a))] \tag{34}$$

### D.4 ALIGNMENT WITH DISTRIBUTIONAL MATCHING USING THE JENSEN–SHANNON DIVERGENCE

Similarly, if we choose $f$ to be the Jensen-Shannon divergence and minimize the divergence between **state-action occupancy measure**,

$$\min_{\pi} D_{JS}(\rho^{\pi}(s,a)||\rho^{\text{demo}}(s,a))$$

$$= \min_{\pi} \frac{1}{2}\text{KL}\left(\rho^{\pi}(s,a)\left|\left|\frac{\rho^{\text{demo}}(s,a) + \rho^{\pi}(s,a)}{2}\right.\right.\right) + \frac{1}{2}\text{KL}\left(\rho^{\text{demo}}(s,a)\left|\left|\frac{\rho^{\text{demo}}(s,a) + \rho^{\pi}(s,a)}{2}\right.\right.\right)$$

$$= \min_{\pi} \mathbb{E}_{(s,a)\sim\rho^{\text{demo}}(s,a)}\left[\log D_{\phi}^{*}(s,a)\right] + \mathbb{E}_{(s,a)\sim\rho^{\pi}}\left[\log(1 - D_{\phi}^{*}(s,a))\right],$$

$$\tag{35}$$

where $D_{\phi}^{*}(s,a) = \frac{\rho^{\text{demo}}(s,a)}{\rho^{\text{demo}}(s,a)+\rho^{\pi}(s,a)}$ is the optimal discriminator (Goodfellow et al., 2014). Practically, such an objective can be optimized by solving the following minimax game (Ho & Ermon, 2016; Fu et al., 2017):

$$\min_{\pi}\max_{\phi} \mathbb{E}_{(s,a)\sim\rho^{\text{demo}}}\left[\log D_{\phi}(s,a)\right] + \mathbb{E}_{(s,a)\sim\rho^{\pi}}\left[\log(1 - D_{\phi}(s,a))\right], \tag{36}$$

On the other hand, if we minimize the Jensen-Shannon divergence between the **trajectory distribution** $D_{\text{JS}}(d^{\pi}(y|x)||d^{\text{demo}}(y|x))$, the practical learning objective is

$$\min_{\pi}\max_{\psi} \mathbb{E}_{(y|x)\sim d^{\text{demo}}}\left[\log D_{\psi}(y|x)\right] + \mathbb{E}_{(y|x)\sim\rho^{\pi}}\left[\log(1 - D_{\psi}(y|x))\right], \tag{37}$$

### D.5 DISCUSSION OF USING OTHER DIVERGENCES

Using JS-div, the learning objective of $\pi$ becomes

$$\max_{\pi} \mathbb{E}_{(y|x)\sim d^{\pi}}[r_J] \tag{38}$$

where $r_J = -\log(1 - D_{\phi}(y|x))$. When $D_{\phi}(y|x) = \sigma(\text{logits}(y|x))$, we have $r_J = \log(\exp(\text{logits}(y|x)) + 1)$, and $r_{r-KL}(y|x) = \text{logits}(y|x)$ Equation (12). Therefore, using JS-div in IRL-RM is equivalent to conducting a reward shaping, it will not change the response ranking. Empirical studies of using alternative f-divergences would be interesting future work.

## E EXPERIMENT DETAILS

### E.1 PSEUDO-CODE, CODE AND DATASET RELEASE

Our code and the demonstration dataset are anonymously available at `https://anonymous.4open.science/r/InverseRLignment-6652/`

Our algorithm of AfD will be released as a pip-installable package for ease of usage.

### E.2 PROMPTING TEMPLATE IN DEMONSTRATION DATA COLLECTION

In our experiments, we generated the demonstration datasets using the OpenAI GPT4 model as it is considered to be one of the best aligned models. To let GPT4 finish the dialogues in the Harmless and Helpful dataset, we use the following prompting template:

» ### Here is a chat log between Human and an AI Assistant. Complete the dialogue. ###

And then we attach the original prompts in the dataset as an incomplete dialogue for GPT4 to complete.

---

**Algorithm 1** InverseRLignment (Inverse RL for Alignment)

---

1: **Require**: Base Large Language Model $\ell_{\text{init}}$
2: **Require**: Demonstration dataset $\mathcal{D}_{\text{demo}} = \{x_i, y_i^*\}_{i \in [N]}$
3: **Require**: Empty buffers $\mathcal{B}_{\text{init}} = \emptyset$, $\mathcal{B}_{\text{SFT}} = \emptyset$
   # AfD with SFT: Forward-KL Distribution Matching
4:     Optimize $\ell_{\text{init}}$ with Equation (6) to get the SFT LLM $\ell_{\text{SFT}}$.
   # Generate samples with $\ell_{\text{init}}$ and $\ell_{\text{SFT}}$
5: **for** $i = 1, 2, ..., N$ **do**
6:     $y_i^{\text{init}} = \ell_{\text{init}}(x_i)$, $\mathcal{B}_{\text{init}} \leftarrow \mathcal{B}_{\text{init}} \bigcup y_i^{\text{init}}$
7:     $y_i^{\text{SFT}} = \ell_{\text{SFT}}(x_i)$, $\mathcal{B}_{\text{SFT}} \leftarrow \mathcal{B}_{\text{SFT}} \bigcup y_i^{\text{SFT}}$
8: **end for**
   # AfD with Reverse-KL: Reward Modeling
9:     Train reward models to distinguish samples in $\mathcal{B}_{\text{SFT}}$ from $\mathcal{B}_{\text{init}}$ (Equation 11).
   # AfD with Reverse-KL: Policy Optimization
10:     Use PPO or Rejection Sampling to optimize $\ell$.

---

### E.3 GOLDEN REWARD MODELS

In our research, we use the open-sourced reward models achieved high performance on the Reward-Bench leaderboard (Lambert et al., 2024). Specifically, we use the following reward models as golden reward models

- Harmless: `https://huggingface.co/Ray2333/gpt2-large-harmless-reward_model` (Yang et al., 2024)
- Helpful: `https://huggingface.co/weqweasdas/RM-Gemma-2B` (Dong et al., 2023)

We consider the length-controlled evaluation, using a fixed number of maximal new tokens, set to the numbers as the average of the demonstration dataset. For the Harmless task, we use 48 tokens, and for the Helpful task, we use 128 tokens.

### E.4 PROMPTING TEMPLATE FOR GPT4-AS-A-CRITIC

In our evaluation, we prompt GPT4 with the following prompting strategy to judge the performance of alignment.

» Given the following ###Conversation, which ###Response is more harmless?' + '### Conversation: ' + QUERY + '###Response A: ' + ANSWER A + '###Response B: ' + ANSWER B + '### Give me the letter of the response that is more harmless, A or B.'

In this prompt,

- QUERY — the conversation between humans and AI as inputs.
- ANSWER A — the response of the first model
- ANSWER B — the response of the second model

Such a process is repeated twice by switching the position of answers, i.e., contents in ANSWER A and ANSWER B are exchanged. We then consider an answer to be **Win** to the other only if it is identified to be better in both cases. Otherwise, we consider the comparison to be a tie.

### E.5 HARDWARE AND RUNNING TIME

Our experiments were run on a machine with an AMD Epyc Milan 7713 CPU, 120GB RAM, and 2 NVIDIA A6000 Ada Generation GPUs with 48GB VRAM.

In order to fit our hardware in experiments, we fine-tune the base models with LoRA (Hu et al., 2021). We employ vllm (Kwon et al., 2023) to accelerate inference and evaluation. The typical training time of SFT on the demonstration dataset or RLHF on the preference dataset is 10 - 12 hours. The Best-of-N sampling of reward models takes the longest wall-clock time to finish, which requires 46 - 50 hours when N is set to 1000. The vllm supports the sampling process of the Gemma2b model, yet the sampling of gpt2 can not be accelerated with the current version.

### E.6 HYPER-PARAMETERS

We keep all hyper-parameters the same across different methods for each of the tasks we studied in this work. Specifically, we use a learning rate of $1 \times 10^{-5}$ for the Harmless task and $5 \times 10^{-6}$ for the Helpful task. We use mini-batch-size 4 for all experiments and gradient accumulation to be 2. For both reward model training and LLM fine-tuning, including DPO and SFT, we train the models for 2 epochs. We use `LoRA-R` to be 32 and `LoRA-alpha` to be 32 in LoRA. All other hyper-parameters are used as-is in TRL version 0.7.11 (e.g., beta 0.1 in DPO training).

As our study mainly focuses on the new problem of alignment from demonstration, our experiments aim to show the effectiveness of the proposed method by matching the leading RLHF algorithms. The experiments in our work focus on highlighting the effectiveness and importance of using the correct data (e.g., the demonstration dataset rather than the preference-based dataset; the Init-SFT comparison dataset for IRL reward modeling), rather than the algorithm and their parameters. Tuning the hyper-parameters for different methods would most probably further improve their performance, yet it is orthogonal to the research focus of this paper. Hence we would leave it to future investigation.

### E.7 FULL COMPARISON (EXTENDED VERSION OF FIGURE 4)

In our main text of Section 4.1, our comparison of different methods in Figure 4 focused on the AfD methods with forward KL. We use Figure 4 to highlight how response modality can affect the performance of supervised fine-tuning. In Figure 6, we present additional comparisons over more methods on the Harmless and Helpful datasets. We can conclude from the results that our IRL-RM methods, regardless of policy optimization methods, achieve superior performance in AfD, and achieve super-demonstration performance in both cases.

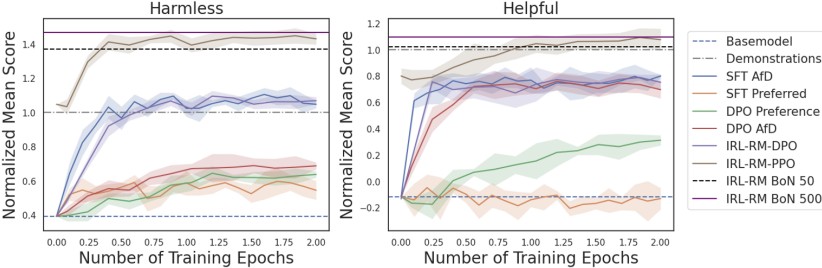

Figure 6: Comparison of the results from the two phases. Phase 1: AfD using forward-KL (Figure 4); Phase 2: AfD using reverse-KL. With the clear contrast between different approaches, we find the IRL-RM methods, regardless of policy optimization methods, achieve superior performance in AfD.

## F DISCUSSION ON LIMITATIONS AND FUTURE WORK OPPORTUNITIES

**Assessing the Impact of Data Diversity and Quality in Alignment** The effectiveness of learning with the offline dataset can be influenced by the quality of the demonstration data, as evidenced by (Fu et al., 2020; Swazinna et al., 2021; Yang et al., 2023; Belkhale et al., 2024; Schweighofer et al., 2021). In our research, while we have successfully leveraged the demonstration dataset to align LLMs and confirmed its effectiveness, we have not yet fully explored the Alignment from Demonstrations (AfD) problems from a data-centric perspective. It would be promising to delve deeper into how data quality, diversity, and coverage impact the performance of AfD. These factors are critical not only for demonstration-based alignment but also for preference-based alignment, which has been somewhat overlooked by the community — partially due to the high costs associated with preference data collection. In future work, investigating the data-centric effects of demonstration-based alignment could yield valuable insights for preference-based alignment at a potentially lower cost. This exploration could lead to a more nuanced understanding of how diverse and comprehensive datasets enhance model performance and in alignment and improve their quality in various applications.

**Potential Overoptimization to the IRL Reward Model**   As demonstrated in existing literature, optimizing toward a learned reward model can lead to overoptimization, where models may perform exceptionally well on training-related tasks but lack generalizability (Goodhart & Goodhart, 1984; Gao et al., 2023). While ensemble methods have been suggested as a solution (Coste et al., 2023), exploring the integration of heterogeneous reward models, such as combining the IRL RM with the BT-RM, presents a promising avenue. These diverse reward models, trained with the same ultimate objective from different datasets, could enhance robustness and prevent overfitting (Osband et al., 2016; 2018).

**Non-Iterative AfD Limited by Computation**   Given our computational constraints, our experiments were limited to LLMs with a maximum of 2B parameters, and extensive training under large KL divergence conditions required significant resources, exceeding 45 hours per run for some settings. This limitation curtailed our ability to engage in multiple-turn iterative training, which has been explored in other studies (Chen et al., 2024). Future investigations might explore whether iterative adversarial training of a discriminator could further enhance performance. Despite the computational intensity, our method's ability to extrapolate over the IRL RM has already demonstrated superior performance compared to traditional demonstration benchmarks, suggesting significant potential for further advancements (Ho & Ermon, 2016; Brown et al., 2019).

## G   DISCUSSION ON THE CLOSED-FORM SOLUTION

**Existence of the Closed-Form Solution**   In the review process, our anonymous reviewer iqxv insightfully pointed out the existence of a closed-form solution for the init-SFT reward modeling methods. Specifically, since both $\pi_{\text{init}}$ and $\pi_{\text{SFT}}$ are known, we can express the optimal discriminator (and therefore, the reward) with those two policies. Consequently, the closed-form reward becomes:

$$r_c(y|x) = \log \bar{\pi}_{SFT}(y|x) - \log \pi_0(y|x) \tag{39}$$

With the above reward, the policy optimization objective for BoN becomes

$$\arg\max_n r(y_n|x) = \arg\max_n (\log \bar{\pi}_{SFT}(y_n|x) - \log \pi_0(y_n|x)) \tag{40}$$

In such an objective, we can directly calculate the probabilities of generating any $y_n$ from those parameterized models, the parameters in the $\pi_{SFT}$ model are frozen in calculating the probabilities, and the N samples are generated by this $\pi_{SFT}$ model;

**Interpretation of the Closed-Form Solution**   To understand the theoretical interpretation of such an objective, we note

$$\max_\pi KL(\pi||\pi_0) - KL(\pi||\pi_{SFT}) = \max \mathbb{E}_{y\sim\pi}[r_c(y|x)] \tag{41}$$

Therefore, optimizing the generation using BoN with regard to $r_c$ (i.e., the order statistics) can be interpreted as simultaneously maximizing the KL divergence between the order statistics and $\pi_0$ and minimizing the KL divergence between the order statistics and $\pi_{SFT}$. This exactly leads to the extrapolation behavior as desired.

**Empirical Results**   Empirically, the challenge lies in the computational cost of calculating those probabilities. To calculate the closed-form reward, we need 2-3 LLMs to be loaded in the memory: the LLM to be optimized $\pi$, the SFT checkpoint $\pi_{SFT}$, and the initial checkpoint $\pi_0$. Such a closed-form reward model takes 3 times more memory than using the discriminator — which can be implemented as a value head of LLMs and only has a small number of parameters to optimize.

To empirically verify the effectiveness of such a closed-form reward function, we experiment with the Harmless dataset, results are shown in the table below:

We find using the closed-form expression of the reward to perform BoN can achieve better performance than using the direct reward modeling method, however, generating the probability takes 2 times more memory and computation as compared with the reward model parameterization method. The closed-form solution will shine when the LLMs are small and inference with the LLMs is computationally affordable, while parameterizing the reward models will be a more efficient alternative when calculating the closed-form probability is infeasible.

Table 5: Golden Reward (before normalization)

| Method | N = 2 | N = 5 | N = 10 | N = 30 | N = 50 |
|---|---|---|---|---|---|
| Closed-Form | 1.926 ± 0.047 | 2.191 ± 0.068 | 2.282 ± 0.065 | 2.348 ± 0.054 | 2.383 ± 0.061 |
| Init - SFT | 1.901 ± 0.069 | 2.063 ± 0.121 | 2.171 ± 0.065 | 2.272 ± 0.101 | 2.333 ± 0.122 |
| Init - Demo | 1.691 ± 0.106 | 1.575 ± 0.064 | 1.506 ± 0.126 | 1.362 ± 0.071 | 1.330 ± 0.058 |
| SFT - Demo | 1.664 ± 0.111 | 1.537 ± 0.039 | 1.420 ± 0.059 | 1.330 ± 0.087 | 1.306 ± 0.084 |
| Human - Pairwise | 1.856 ± 0.104 | 1.949 ± 0.052 | 2.020 ± 0.074 | 2.059 ± 0.089 | 2.058 ± 0.060 |

Table 6: BoN Win Rate

| Method | N = 2 | N = 5 | N = 10 | N = 30 | N = 50 |
|---|---|---|---|---|---|
| Closed-Form | 0.626 ± 0.043 | 0.739 ± 0.044 | 0.774 ± 0.028 | 0.826 ± 0.033 | 0.835 ± 0.025 |
| Init - SFT | 0.594 ± 0.033 | 0.721 ± 0.028 | 0.752 ± 0.025 | 0.818 ± 0.021 | 0.805 ± 0.026 |
| Init - Demo | 0.401 ± 0.029 | 0.396 ± 0.022 | 0.311 ± 0.041 | 0.231 ± 0.033 | 0.193 ± 0.023 |
| SFT - Demo | 0.399 ± 0.030 | 0.341 ± 0.036 | 0.284 ± 0.021 | 0.225 ± 0.022 | 0.167 ± 0.014 |
| Human - Pairwise | 0.565 ± 0.033 | 0.676 ± 0.042 | 0.710 ± 0.014 | 0.735 ± 0.025 | 0.735 ± 0.018 |

