# OpenReview forum: "Large Language Model Alignment via Inverse Reinforcement Learning from Demonstrations"
_ICLR.cc/2025/Conference — Submitted to ICLR 2025_

### Official Review · Reviewer_wScp · 2024-11-01

**Soundness:** 2
**Presentation:** 2
**Contribution:** 1
**Rating:** 5
**Confidence:** 4

**Summary:**

This paper presents AfD, a sort of framework for learning from demonstrations in LLMs. The authors do a number of different experimenst within this framework on a number of different methods.

**Strengths:**

* The work attempts to unify a number of diverse ideas, which is helpful
* The work makes nice use of different colored boxes so things can easily be found.

**Weaknesses:**

**Novelty**
I am unsure what exactly is novel in this work. To my knowledge nothing the authors introduce is explicitly new, or has new experiments.
* Sec 2.2: This MDP breakdown for LLMs is well known
* Sec 3.1: It is well known that SFT = BC
* Sec 3.1: I have not looked into the descriminator objective to see if it is in prior work, but the authors don't use it in experiments.
* Sec 3.2: The idea of using the model generations as negatives is done in SPIN and in DITTO (Show don't tell, Shaikh et al.) DITTO also does something similar to this paper by SFTing the model first before sampling.
* Sec 4.1: These experiments show SFT > RLHF on the same number of demos. I don't find this surprising, similar results are also in Shaikh et al.
* Sec 4.2: I think the section may be where the authors find novelty?

Overall, the paper seems to focus a lot on unifying different ideas that have existed for a while. While this is OK, the paper is not written as if it were a survey and at present it sounds like the authors are claiming AfD to be some new framework that has not been extensively studied before.

**Writing**
The paper is a bit hard to follow since there are so many subjects. I was initially confused as to what was being evaluated in each experimental sesction. For example, it was initially unclear to me what the different baselines were in Sec 4.1.

**Experiments**
* the experimental results at present do not seem compelling.
* Sec 4.1: It makes sense that SFT with demos does better than RLHF. The amount of data isn't reported on however, and so its unclear what the cost of data collection vs performance tradeoff is.

**Missing Citations**
This work brings together a lot of different ideas, which is great, but the authors seem to miss a ton of related work which has already covered very similar ideas:

* IRL: Ziebart is the OG in maxEnt IRL.
* From r to Q* by Rafailov et al. - Token level MDP
* Show don't tell: Aligning LLMs with demonstrated feedback by Shaikh et al has very similar ideas
* Preference Fine-Tuning of LLMs Should Leverage Suboptimal, On-Policy Data by Tajwar et al. covers mode seeking behavior.
* Imitating language via scalable inverse reinforcement learning by Wulfmeier et al for IRL on LLMs


## Recommendations
I would recommend that for a future draft the authors either a) refocus the draft to be a survey on applying concepts traditionally used in IRL to language models or b) focus on the reward modeling experiments.

**Questions:**

Can the authors summarize the main contribution of the work? Is there something I am missing?

---

> ### Author Response · Authors · 2024-11-18
> **Author Response to Reviewer wScp (Part 1)**
>
> We sincerely thank the reviewer for their time and thoughtful feedback. We value the opportunity to clarify the novelty, contributions, and focus of our work, as well as address any concerns raised. Below, we provide detailed responses to the key points.
>
> ---
>
> ## 1. Responses Regarding Novelty
>
> **We respectfully disagree with the assessment that our work lacks novelty. Importantly, our paper predates several of the works mentioned by the reviewer.**
>
> ---
> **(1). Timeliness:**
>
> To the best of our knowledge, before the submission deadline of ICLR, there were only three papers on this topic, they were made publicly available later than our work. We are happy to provide additional evidence by compliant with ICLR policies.
> - Our paper is 6 months earlier than, has been acknowledged, and cited in Wulfemier et al. (Sep. 2024), as their prior related work.
> - Our paper is 3 months earlier than DITTO, which is currently **a concurrent submission to ICLR 2025.**
> - Our paper is 1 month earlier than Tajwar et al. And the paper focused on sub-optimal, on-policy data in RLHF.
> - Our paper is 1 month earlier than Rafailov et al. The problem **we characterized with MDP in our paper is more general than theirs** — AfD as IRL is different from the conventional RLHF setup.
>
> We would like to emphasize that the ideas introduced in our paper were developed and documented well before the appearance of these works. We believe this timing further supports the originality and relevance of our contributions.
>
> ---
> **(2). Discussions on Ziebart et al. and SPIN in Our Paper**
>
> In our submitted manuscript, we have appropriately cited and acknowledged relevant prior works, including those by Ziebart, which lays the foundation for our approach.
>
> On SPIN, we **had a section discussing SPIN as an important related work. In our initial submission, we had a page starting from line 1003 to line 1058 discussing the link and difference between our work and SPIN**. As per suggested and acknowledged by reviewer iqxv that this section itself has a unique contribution to the community, we have moved this section to the main text in our revision (lines 316 - 418, highlighted by blue text). Please refer to our updated manuscript for the revision!
>
> ---
> **(3). Distinct Focus and Contribution:**
>
> While some relevant ideas have been later on discussed and empirically studied in related works, our approach differs significantly. Those related works can serve as additional support for our key insights — as the reviewer pointed out — some of their experiment results shared similar discoveries with ours.
>
> Moreover, the focus and techniques used in those papers are different. The focus of DITTO is primarily on **individualized and few-shot alignment using demonstrations**, whereas our work takes a broader approach, addressing general alignment from demonstration setups. This distinction marks an important difference in the scope and application of the methods.
>
> Additionally, our work introduces a two-stage explicit reward modeling method, which is distinct from the direct alignment methods discussed in SPIN, DITTO, and Wulfmeier et al. We believe that exploring both direct alignment and reward modeling approaches, as we do in our paper, provides a meaningful contribution to the field. The existence of multiple approaches in this area should not be viewed as competing but as complementary solutions that advance the field in different ways.
>
> ---
> **(4). From IRL Foundation to LLM Alignment with IRL**
>
> The reviewer correctly points out that Ziebart laid foundational work, and **we have cited and built upon it.** However, we would like to emphasize that our work **extends this foundational knowledge by proposing novel methods that are specifically tailored for learning from demonstrations in LLMs.** This is not simply a reiteration of past work, but rather an extension and adaptation of existing ideas to a new and impactful context.
>
>
> ---
> **(5). Summary and Request for Re-evaluation**
>
> **We understand the reviewer’s concerns given their knowledge of the recent advancements in the field (but unintentionally missing the factual temporal order!). We hope that the points above help clarify the novelty and contribution of our work.**
> We would also appreciate the reviewer’s recognition of the soundness of our method based on existing empirical studies and agree that it is beneficial for papers in the same domain to support one another in advancing the state of the art.
>
> **We would kindly request reviewer wScp to consider re-evaluating our work, taking into account the above clarifications, and consider the novelty and significance of our contributions in the broader context of the field.**

---

> > ### Author Response · Authors · 2024-11-18
> > **Author Response to Reviewer wScp (Part 2)**
> >
> > ## 2. Clarification of Key Contributions
> >
> >
> > We would like to summarize the key contributions below. First, we would like to list three contributions that were not explored in the **later related works.**
> >
> > 1. Our paper is the first to introduce a **unified framework** for distributional matching that encompasses SFT, RLHF, and AfD (i.e., alignment from demonstration). By framing all these alignment methods through the lens of Inverse RL, our work not only provides a cohesive perspective but also bridges the gap between the domains of IRL and LLM alignment, offering valuable insights into both areas.
> >
> > 2. Technically, in section 3.2, we highlighted why using the models’ generations as negative samples and using demonstrations as positive samples is suboptimal, and how to overcome such a difficulty.
> >
> > 3. With both analysis and empirical results, we showed that explicit reward modeling using the demonstration dataset is the only method that **can achieve super-demonstrator performance.**
> >
> > Moreover, considering the timeline of this line of research, our paper is **factually the first** in:
> >
> > 4. Building reward models and introducing the idea of LLM alignment from the demonstration dataset.
> >
> > 5. Characterizing the general alignment problems using an Inverse RL perspective, and demonstrating how RLHF and AfD are two of its instantiations falling to this general class.
> >
> > 6. Empirically showing the effectiveness of the IRL-based methods in LLM alignment, providing an alternative approach to the conventional preference-based alignment methods.
> >
> >
> > ----
> > ## 3. Response Regarding Experiments
> >
> > (1). Our Results Demonstrate **Statistical Significance** in Validating the Proposed Method.
> >
> > We **respectfully disagree** with the comment that our _"experimental results at present do not seem compelling"_. Our experiments demonstrate the effectiveness of the proposed methods through both theoretical insights and comprehensive evaluations:
> >
> > - In our experiments, we highlighted the superiority of the proposed method with **extensive ablation studies** and demonstrated the effectiveness of building reward models from demonstration datasets. The proposed method **achieves super-demonstrator performance on all experiment setups, verifying the key insight of our work**.
> > - We theoretically and empirically highlighted such a super-demonstrator performance can not be achieved by prior work such as SPIN, due to the lack of reward model extrapolation.
> >
> > (2). Focusing on Reward Model Experiments
> >
> > We appreciate the reviewer’s suggestion to focus on the reward modeling experiments, as **this aligns closely with the primary focus of our experimental design.**
> >
> > In our work, most of our experiment sections **have centered on evaluating reward models** generated from the demonstration dataset. May we kindly reiterate what has been confirmed in our experimental section:
> > - In 4.2, we evaluated different reward modeling choices using the golden reward model evaluation. We find the proposed method achieves the best performance when using the demonstration dataset, and achieves on-par performance than training reward models from preference-based annotations.
> > - In 4.3, we evaluated our reward models using GPT4-as-a-judge evaluation, to provide an additional metric in addition to the golden reward model evaluation and more comprehensive and reliable results that verify the effectiveness of our reward models.
> >
> >
> >
> > ------
> >
> > Once again, we sincerely thank the reviewer for their thoughtful feedback and appreciate the opportunity to provide additional clarifications. **We believe there may have been some misunderstandings and would respectfully request a re-evaluation of our manuscript.** If any concerns remain, please do not hesitate to let us know, and we will do our utmost to address them!

---

> ### Comment · Reviewer_wScp · 2024-11-25
> **Response to authors**
>
> I would like to thank the reviewer for their detailed responses to my questions. I have read the paper again and provide both a) responses to the author’s rebuttal and b) a new review of the work and an updated score.
>
> ## Response to authors
>
> **On Novelty**
>
> *Timeliness*:
> I apologize, my first response may have placed an over-importance on the relationship between the timing of prior work and this current paper. That being said, I am unable to reproduce the timelines given by the authors.
> * I have placed r to Q* at Apr 18, 2024: making the authors work a month later, not a month earlier.
> * I have placed DITTO at June 2nd, making the authors work only a handful of days earlier, not 3 months after
> * I place Tajwar et al. at Apr 22, making the authors work a month later, not a month earlier.
>
> However, I don’t consider the exact timelines here as a crucial component of my assessment of the work. I simply meant to show that several ideas present in this paper were published in work prior work, though not in the exact same form.
>
> There are even other earlier works that do Inverse-RL in the token level MDP, like:
>  Sequence Match: Imitation learning for autoregressive sequence modeling by Cundy and Ermon in 2023 uses Inverse RL in the token-level MDP.
> ON-POLICY DISTILLATION OF LANGUAGE MODELS: LEARNING FROM SELF-GENERATED MISTAKES by Agarwal et al. which was published at ICLR 2024.
>
> *Discussion of SPIN*
> Thanks for bringing this to my attention. I did not read this initially as it was in the appendix. I understand that DPO makes assumptions about the data, and that the authors approach uses a different model than SPIN to generate negatives.
>
> *Distinct Focus and Contribution*
> I understand that DITTO is focused on few-shot alignment, and the SPIN / Wulfmeier et al is focus on SFT or alignment. This response left me with some questions, as it is just stated that this work is designed to address “general alignment from demonstrations” which I assume is just learning from demonstrations. This is a big area, and after re-reading the paper I believe some of the author's contributions are new, and some are not. I’ll discuss this more later.
>
> *From IRL Foundation to LLM Alignment*
> I agree that the authors have presented new ideas in their work. I just think a large part of the work as it is currently is focused on re-iterating that we can consider each step of generation an MDP, that forward KL = BC and that reverse KL = IRL which can be optimized adversarially. I am interested in the authors new experiments on how to optimize these objectives, and ablations over the way in which the discriminator is learned. I wish the paper focused on these as the contribution, which I believe are novel and useful, rather than the general framework.
>
> **Clarifications of Key Contributions**
>
> 1. Framework: This unified framework for distribution matching has existed for a long time outside of LLMs, and works in LLMs have already used the unified framework, even though it was not explicitly written up.
> 2. Sec 3.2: I believe this is interesting and is new contribution. I wish the authors spent more time discussing this, and made this more of a core contribution of the work. However, section 3.2 begins only on page 6. I think the work would be much improved if it focused primarily on how to effectively learn a reward function from demonstrations for inverse RL, and ablating and explaining these choices.
>
> Regarding timeline, I respectfully disagree on the author's points:
> 4. “The first to introduce the idea of LLM alignment from demonstrations” I think this was done in SeqMatch (f-divergence minimization) and perhaps SPIN which were both definitively published before this work
> 5. “Inverse-RL Perspective” I think this was done by seqmatch and other works which took the distribution matching perspective over occupancy measures, but did not call it InverseRL.
> 6. Previous works have shown IRL is effective, but I don’t believe this diminishes from the authors contributions which show that IRL can be effective! I think instead the authors should re-focus their work around their contributions.
>
> **On Experiments**
>
> * I see that the reviewers get better performance using demonstrations than pairwise preferences. I have further questions and weaknesses about this I detail in my new review of the work.
> * “We theoretically and empirically highlighted super-demonstrator performance”:
> 1. Could the authors kindly point me to where the theoretical evidence of super-demonstrator performance is, and where the proof is that SPIN cannot achieve it.
> 2. Could the authors clarify what they are empirically measuring super demonstrator performance with respect to? For example, doing better than the preferences isn’t evidence of this because the demonstratiosn weren’t generated by the same policy!
>
> *Focus on Reward Model Experiments*
> These are interesting, and I appreciate the authors bringing these up again. I have discussed them in my new review of the work.

---

> ### Comment · Reviewer_wScp · 2024-11-25
> **New Review**
>
> ## New Review
> The authors present AfD, a general framework for alignment using demonstrations. That unifies SFT and inverseRL by looking at different directions of the KL divergence.
>
> **Strengths**
> * Several works have demonstrated that the quality of the reward model is extremely important to performance. I think the authors' experiments address an interesting and important question.
> * I think the experiments in Section 4.2 are very interesting – demonstrating how the choice of which data to use for a reward model impacts down-stream performance.
>
> **Weaknesses**
> * The authors work primarily focuses on their introduced framework for distribution matching in the Token level-MDP. However, 1) this general framework/knowledge has been known for a long time in RL/imitaiton literature, so I am uncertain if this can be claimed as a contribution. 2) even within the landscape of LLMs, several prior works have used the distribution matching framework as justification for their methods, citing literature in imitation learning. This casts doubt as to whether or not the author's proposed unified framework is adding anything. At present, I do not believe the author's unified framework constitutes a novel contribution, as prior works have already used the same ideas, but did not feel it was necessary to present such a framework. I would be happy to see a survey or overview paper on this topic, but I am unsure that ICLR is the best venue for it. I again believe the work would be stronger if the authors focus on the reverse-KL case, and how to best learn a reward model for it. For citations, I refer the author to my response to their rebuttal.
>
> * The authors claim to be doing Inverse-RL with their method in Section 3.2 and Section 4.2, but they learn the reward model / discriminator from a static dataset. Since the inverseRL objective requires the policy to be optimized under its own distribution, its unclear how a static discriminator will actually optimize for the objective the authors claim. This is a bit strange as the authors for most of the work claim that they are introducing a framework for IRL, but then don’t actually seem to do inverse RL in the end.
>
> * The authors perform their experimental evaluation on only a single task. Most contemporary papers have a more extensive evaluation.
>
> * The experimental evaluation raises a number of questions, which I bring up in the “questions” section of my review. I think these questions need to be discussed to gain confidence in the experimental section.
>
> **Questions**
> My questions are largely concerning the experimental evaluation.
>
> * The authors use GPT4 to generate demonstration data. However, in Section 4.1 the authors compare finetuning on data generated by GPT4 (SFT AfD, DPO AfD) to finetuning on data from the original dataset (SFT Preferred, DPO Preference). The only conclusion I can draw from this experiment at present is that the GPT-4 data is better, preferred by the gold RM, or easier to  fit than the original data. Why is comparing between the original data, and data distilled from GPT-4 a valid comparison on this task? Especially because GPT-4 has already likely been trained to be “helpful and harmless”.
>
> * Again, for Section 4.3 the authors use GPT-4 as a critic. However, the BoN IRL-RM and SFT models were trained on data generated by GPT-4, which GPT-4 likely prefers. Is there a reason for why this is a valid comparison? Would GPT-4 not be biased to its own responses, and thus prefer SFT/BoN IRL-RM to the BT-RM.
>
> * Could the authors show experiments that work on another dataset?
>
> * Could the authors clarify what the theoretical and empirical evidence of extrapolation beyond the demonstrator is as this was repeatedly brought up in rebuttal? The proof of this in Brown et al 2019 made specific requirements on the sub-optimality of the demonstrator, ie there were requirements on the demonstrator’s average performance in relation to the true optimal policy.
>
> * Could the authors explain the relationship between the objective that they actually solve for in the experiments (Init-SFT) versus the theoretical section (Reverse-KL occupancy matching). I understand that occupancy matching can be done with a discriminator/RM, but the distribution on which the discriminator is trained is important.
>
> **Concluding Thoughts**
> I still think that a paper which focused on the best way to learn a discriminator from demonstrations would be impactful and insightful. However, at present the work spends a lot of time / focus on other parts, and in my opinion as a result does not give the “learning a discriminator” part proper treatment. Moreover, I still have questions regarding the experimental evaluation which underpins this, as the authors base many of their conclusions off comparisons between models trained on GPT-4 demonstrations and models trained on the original HH data. This seems like it might be problematic, particularly when using GPT-4 as a critic.

---

> ### Author Response · Authors · 2024-11-25
> **Author Response to the New Review (Part 1)**
>
> We sincerely thank reviewer wScp for their consideration of re-evaluating our paper!
>
> Although the author response window is narrowing down and our response may be limited by the approaching deadline, we deeply appreciate such an opportunity to further clarify the contribution and a few other aspects of our work. In the following, we would respond to the reviewer’s comments grouped by topics.
>
> ----
>
> ## 1. Motivations of introducing the framework before the technical method.
>
> We thank our reviewer for their affirmative comment on our method section (Sec. 3.2)! We agree the explicit reward modeling part, together with the technique we introduced to overcome the reward hacking challenge, are the major technical contributions of our work. However, we would like to respectfully argue the current presentation design of the paper is motivated by the following considerations:
>
> (1). For a self-contained paper.
>
> Providing the prerequisite knowledge of Inverse RL and the MDP formulation is necessary for the follow-up discussion of e.g., distribution matching in our paper.
>
> We understand those prerequisites might be straightforward to our knowledgable reviewers, yet for our general potential readers who are not familiar with MDPs and RL, our work is self-contained such that any reader familiar with the basic concept of LLMs to be able to understand our paper.
>
> (2). Connecting research areas and inspiring future innovations.
>
> Bridging the research area of Inverse RL and Imitation learning with the general task of alignment — including both alignment from preference feedback and demonstrative feedback is useful to the community. While the idea is not totally new by itself and has been discussed in the pre-LLM era (e.g., as we have discussed in the related work section, and pointed out by the reviewer), we believe it is beneficial to highlight its importance in the LLM alignment setups. It is worth letting more people understand the connection between RLHF and AfD through the lens of Inverse RL: **RLHF and AfD are two different instantiations of Inverse RL for LLM alignment, the difference is in how to generate the reward signal**.
>
> We would highlight such a unified framework can inspire potential future innovations beyond RLHF or AfD — the essential idea is not to use a specific data type for alignment, but more on leveraging different types of datasets to build the reward model. The current focus of the literature mainly focuses on RLHF — using preference-based feedback — except for the few recent exceptions (concurrent work) pointed out by our reviewer.
>
>
> (3). Disclosing the motivation of the proposed method.
>
> Describing the token generation process using the MDP language is essential because it can be used to formally define the _forward and inverse process_. Within such a framework, we are able to formally discuss the alignment problem using the RL and Inverse RL perspective, introduce the distribution matching methods that have been applied in the literature, and adapt those methods according to the property of the AfD task (Sec. 3.2)

---

> ### Author Response · Authors · 2024-11-25
> **Author Response to the New Review (Part 2)**
>
> ## 2. Why explicit reward modeling can be superior to implicit methods (e.g., SPIN, SFT)
>
> (1). On the sub-optimality of demonstration data
>
> We fully agree and are glad to see the reviewer resonates with our high-level insight in reward extrapolation (cf. Brown et al. 2019). The reviewer is very correct in pointing out if the demonstrations are **optimal**, then there would be no further space for improvement.
>
> In the context of LLM alignment, even the responses generated by GPT4 are far from **optimal**. On one hand, this is because the concept of helpfulness or harmless themselves may not be transitive. On the other hand, GPT4 will reject to give answer to many queries in the harmless dataset due to its filtering mechanism. Therefore, the key insights of Brown et al. 2019 that extrapolating over reward models can improve over **sub-optimal** demonstration can be applied in our context.
>
> In our previous response and also in our paper, we highlighted the core difference between the direct alignment methods without a reward model, such as SPIN, which is upper bounded by the demonstration quality, because they always consider the demonstration data to be positive, and the **current generation** as negative.
>
> This can be proven by contradiction: assuming in SPIN the LLM to be aligned outperforms the expert demonstration, then, in the next generation, the algorithm will treat those (better) generations as **negative samples** and optimize the policy to decrease the probability of generating such responses — this contradicts with the objective of increasing the probability of generating higher-quality responses.
>
>
> (2). Policy optimization objective with our explicit reward models
>
> Define
>
> $$r_{c}(y|x)=\log \bar{\pi}_{SFT}(y|x) - \log \pi_0 (y|x)$$
>
> The policy optimization objective with our explicit reward model becomes
>
> $$\arg\max_{n} r(y_n|x) = \arg\max_{n}(\log\bar{\pi}_{SFT}(y_n|x) -\log \pi_0(y_n|x) )$$
>
> In such an objective, we can directly calculate the probabilities of generating any $y_n$ from those parameterized models, the parameters in the $\pi_{SFT}$ model are frozen in calculating the probabilities, and the N samples are generated by this $\pi_{SFT}$ model;
>
> To understand the **theoretical interpretation of such an objective**, we note
>
> $\max_\pi KL(\pi||\pi_0)-KL(\pi||\pi_{SFT})=\max \mathbb{E}_{y\sim\pi}[r_c(y|x)]$
>
> Therefore, optimizing the generation using BoN with regard to $r_c$ (i.e., the order statistics) can be interpreted as simultaneously maximizing the KL divergence between the order statistics and $\pi_0$ and minimizing the KL divergence between the order statistics and $\pi_{SFT}$. **This exactly leads to the extrapolation behavior as we desired!**
>
> ----
> ## 3. On evaluations.
>
> (1). The pursuance of a comprehensive comparison in our work.
>
> Firstly, we would like to highlight the efforts made in our work to make the results reliable — we have used both golden-reward evaluation and GPT4-as-a-judge to provide a comprehensive evaluation of the proposed methods. Our evaluation strictly follows the best practices in the literature.
>
> (2). Motivation for using the HH Dataset (Harmless and Helpful tasks)
>
> Please kindly let us reiterate that the most important motivation for using the Harmless and Helpful datasets is that the open-sourced golden reward models exist on those tasks, and those tasks are well-studied in the literature.
>
> On other datasets such as UltraFeedback, policy evaluation can only be based on GPT4-as-a-judge, which is much more expensive than using open-sourced golden reward models and may suffer from the challenge of why is GPT4 able to do the evaluation. Should the reviewer agree with us among many literature that GPT4-based evaluation is reliable, we are happy to run experiments on UltraFeedback (which can be costly).
>
> (3). Can GPT4 evaluations be trusted?
>
> First of all, we would like to highlight the fact that our evaluation is based on 2 methods, the effectiveness of the proposed method is verified by both evaluation metrics.
>
> “the BoN IRL-RM and SFT models were trained on data generated by GPT4, which GPT4 likely prefers.” **We would respectfully disagree with the reviewer on this ungrounded claim** — the base models we used are not GPT4, and their generated contents are different from GPT4. More importantly, we would like to highlight that the objective and takeaway of Section 4.3 is about the **relative improvement** achieved from SFT to Inverse RL, rather than the **absolute performance**.
> Therefore, the takeaway of Section 4.3 is isolated from whether or not GPT4 would prefer the contents generated by AfD on some specific demonstrations — as existing literature using GPT4 evaluation did.
>
>
> -----
>
> Once again, we sincerely thank the reviewer for taking the time to review our paper. Despite the limited time remaining, we are still eager to address any further concerns or questions you may have.

---

> ### Comment · Reviewer_wScp · 2024-11-25
> **Response to the authors**
>
> I would like to thank the authors for their response. I found the clarificaitons helpful, but think we are still not aligned on a few points.
>
> **Motivations of the Framework**
> I agree with the authors that some background is necessary for the paper. However, my issues with this part of the paper remains as follows:
> 1. The authors posit in their response and the paper text that this framework is a new contribution. I think we disagree on this point.
> 2. The length of discussion on the framework leaves less time to discuss the later contributions in text.
> 3. The presented framework is not well connected with the method the authors propose later in the paper. For example, in their response the authors detail the objective that the experiments optimize for, which is not exactly the reverse-KL. Why is this objective not in the main text of the paper? I think there should be a discussion in the main text.
>
>
> **On Explicit Reward Model**
> I agree with the point that explicit reward modeling can be better. I also thank the authors for detailing the proof by contradiction, and providing the objective. I believe these should be more central components of the paper, as at present they are not reflected within the text.
>
> However, I think a few points are missing from the paper:
> * The authors should be explicit that they are only providing a proof of contradiction that SPIN cannot achieve super demonstrator performance. I had the impression that there was a theoretical proof that the authors method could achieve super-demonstrator performance. While empirically this seems to be the case, the distinction between theoretical and empirical results should be clear.
> * the steps to go from reverse KL to the authors objective should be in the paper (I think I see how it is done)
>
>
> **Evaluations**
>
> 1) The distinction I am making is that while other works use GPT-4 as a critic, they do not necessarily use GPT-4 to generate the data. A more grounded approach in my mind would be to take a demonstration / SFT dataset, and label preferences between them. Then the demonstrations and preferences come from the same distribution, and one is not biased towards the evaluator. There have been several works (including DITTO) which note the bias of GPT-4 as a critic when judging its own outputs.
>
> 2) Other datasets: the authors might consider showing their results on other common datasets used for SFT, or even datasets in preference based learning, like TLDR or even IMDB sentiment. Again, the key distinction I am making is that the authors experiments use GPT-4 to generate the demonstrations, then use GPT-4 as a critic. I am arguing that using demonstrations that are not generated by GPT-4 would be more reliable when evaluating with GPT-4.
>
> 3) I might be mis-understanding the draft. However, I was basing this discussion off of this statement: “Demonstrations were generated using the OpenAI GPT-4 API, with detailed prompting strategies…” where the prompting strategy inputs the dataset and asks GPT-4 to complete it. Then, the AfD models use these GPT-4 demonstrations for reward learning. In this sense, the AfD models are trained on data generated by the critic, while the baselines are not. Please correct me if I am wrong.
>
>
> **Overall Recommendation**
> I agree with many of the points the authors bring up in their responses, and find the responses very useful. My recommendation remains that the authors re-vamp the writing of the paper to focus less on the “framework” such that their method can be adequately explained within the main text of the paper. Several details and contributions discussed here would make the main body of the paper much stronger, but are omitted because of space.

---

> > ### Author Response · Authors · 2024-11-29
> > **Author Response / Adding Requested Experiments**
> >
> > Thank you for the further feedback!
> >
> > ### 1. On Necessity of Framework
> >
> > We thank our reviewer for their affirmative comments on our contributions!
> > We see the reviewer’s point and appreciate their comments. In our current draft, we had pages 1-2 for the introduction (including Figure 1, the roadmap of the paper), pages 3-4 for the framework (as preliminaries), pages 5-8 for the method, and finally pages 9-10 for the experiments. Such an arrangement on different has received acknowledgment from Reviewer iqxv (rated as excellent).
> >
> > Please kindly permit us to respectfully maintain a different presentation strategy :)
> >
> >
> > ### 2. On Explicit Reward Model
> >
> > We thank the reviewer for their further feedback on the interpretation of the reward model. In our latest manuscript, we have included a detailed discussion on closed-form solutions in Appendix G. Please refer to page 28 in our updated manuscript for the details. We’ve highlighted the new content with blue text.
> >
> >
> > ### 3. On Evaluations
> >
> > We thank our reviewer for their further explanation. We would like to use the following new experiment results to address the reviewer’s concerns.
> >
> > We thank the reviewer for pointing out the insightful discoveries in DITTO. **However, we would respectfully note that DITTO is another ICLR submission. Requesting one ICLR submission to follow the experiment setup of another submission may break the ICLR code of conduct.**
> >
> > That being said, we would like to use the following setups to address the reviewer’s concern regarding experiments:
> >
> >
> > **(1). AfD with GPT3.5 Demonstration**
> >
> >  **We additionally experiment with a demonstration dataset generated by GPT3.5.** With multiple datasets and multiple methods studied, we can isolate and understand the specific contributions of demonstration data source, and in this case, the evaluation is based on the **Golden Reward Models.**
> >
> > |Model|Harmless-GPT4|Harmless-GPT3.5|Helpful-GPT4|Helpful-GPT3.5|
> > |-|-|-|-|-|
> > |Demo|1.704|1.615|0.735|0.520|
> > |Base|0.670|0.670|-0.086|-0.086|
> > |SFT|1.785|1.667|0.588|0.433|
> > |IRL-RM (N=10)|2.171|1.755|0.598|0.454|
> > |IRL-RM (N=30)|2.272|1.842|0.692|0.498|
> > |IRL-RM (N=50)|2.333|1.889|0.751|0.537|
> >
> > In this experiment, we studied the effect of the demonstration dataset and showed the effectiveness of the proposed method using Golden Reward Model evaluations.
> >
> > **(2). MT-Bench Evaluation with UltraFeedback**
> >
> > We **added new experiments using the UltraFeedback dataset and evaluated different approaches using the MT-Bench**. Limited by computational resources, we use Gemma-2b and 4-bit quantized Gemma-7b and LoRA in training. Among the multiple responses in the UltraFeedback dataset, we use the **highest-rewarded responses** (rather than generating them by GPT4) as the demonstration dataset to perform AfD.
> >
> > |Method|Gemma-2b|Gemma-7b (4-bit)|
> > |-|-|-|
> > |Base|1.394|2.617|
> > |SFT|1.875|3.250|
> > |DPO (demo)|1.903|3.134|
> > |IRL-RM (N=10)|2.075|3.421|
> > |IRL-RM (N=30)|2.450|3.625|
> > |IRL-RM (N=50)|2.656|3.731|
> > |IRL-RM (N=300)|2.869|4.076|
> >
> >
> > In this experiment, we use **different models for expert demonstration generation and evaluation.**
> > The results presented in the table above indicate that the proposed method significantly outperforms both SFT in the AfD setting and the naive application of DPO --- using the initial policy generation as negative samples.
> >
> >
> > ----
> >
> > We thank our reviewer again for their suggestions for improving our paper. Should there be any leftover concerns, please let us know and we are eager to do our utmost to address them!

---

> > > ### Comment · Reviewer_wScp · 2024-11-30
> > > **Thanks for the additional clarifications**
> > >
> > > I would like to thank the authors for their additional clarifications and experiments -- I have raised my score accordingly.
> > >
> > > I still believe that to make the paper stronger, the majority of the writing in the main paper should focus on the choice of data when learning the reward model (what I believe is the central novel contribution here). Adding additional content to the appendix, while useful, does not address my core concern around the presentation. For ex: the introduction and framework section mostly focus on introducing imitation learning from demonstrations as a new idea for LLMs. The point I would instead focus on is which data to use for learning the reward model from demos.

---

> ### Public Comment · ~Hao_Sun1 · 2025-11-27
> **The peer-review system depends on honesty, not strategic obstruction. Such behavior harms the entire community.**
>
> It is disappointing to observe a reviewer making claims that contradict the actual content of the paper, especially while having a closely related submission of their own.
> Intentional misrepresentation undermines the integrity of the review process.
> The community deserves better.

---

### Official Review · Reviewer_iqxv · 2024-11-03

**Soundness:** 3
**Presentation:** 4
**Contribution:** 2
**Rating:** 5
**Confidence:** 5

**Summary:**

The authors cast LLM alignment as an imitation learning problem, opening up the possibility of learning from demonstrations alone and leveraging insights from inverse RL.

**Strengths:**

- This is an exceptionally well-written paper with crystal-clear exposition and take-aways -- kudos to the authors!

**Weaknesses:**

- (Minor) RLHF is usually framed as KL-constrained /MaxEnt RL, rather than standard RL problem formulation in Eq. 2.

- (Minor) Another good citation for intransitive preferences in RLHF might be https://arxiv.org/abs/2401.04056.

- I would argue that the fact that SFT is BC is fairly well known. It also doesn't seem that surprising that doing SFT on data generated by a super high quality model works well -- the question is of course how we train such a powerful supervisor model in the first place, for which preference data still appears to be neccesary. So, it's hard for me to give many points for novelty for that section of the paper.

- For the most preferred RM strategy (comparing $\pi_{SFT}$ to $\pi_{init}$) , we know the optimal discriminator in closed form -- it is precisely $d^{\star}(x, y) = \log \pi_{SFT}(y|x) - \log \pi_{init}(y|x)$ (if a logistic loss is used, otherwise could be the density ratio in Eq. 9). I don't see the added value in actually learning a separate discriminator for the best-of-N sampling procedure -- it seems like we could only do worse rather than using the log ratio.

- It is a bit disappointing that the final policy requires a BoN step -- I would have liked to see the results of proper policy optimization procedure on the learned RMs.

**Questions:**

1. If it is computationally feasible, could you compare to the closed form for the optimal discriminator in your BoN experiments?

2. If I am understanding correctly, if you used the "golden" RM for BoN, you'd get a win rate of 1?

3. Also, is the model you're sampling from here just the result of SFT on the demos from $\pi_{\beta}$, aka $\pi_{SFT}$? If so, is their a theoretical interpretation of the effect of the BoN procedure with the "closed form" discriminator I mention above?

4. Could you provide more explanation for why the win rate goes down with higher N for several lines in figure 4?

5. If you have space, could you move up the comparison to SPIN to the main paper? I think it is quite interesting and under-appreciated in the broader community -- I have struggled to convince people of precisely the point you are making.

---

> ### Author Response · Authors · 2024-11-18
> **Author Response to Reviewer iqxv (Part 1)**
>
> We thank the reviewer for their encouraging comments on our presentation and for their insightful comments. We would respond to each of the concerns and questions in turn:
>
> ---
>
> ## 1. MaxEnt RL / KL Constraints
>
> Thank you for highlighting this point! In Section 2.2, we intentionally omitted the notion of KL constraints to maintain conciseness and simplify the notation. KL constraints or MaxEnt regularization can indeed be interpreted as an additional (omitted) objective beyond alignment in this context.
>
> In this work, we avoided incorporating KL terms in both entropy constraints and alignment objectives to minimize potential confusion for readers. However, integrating our method and framework into a MaxEnt/KL-regularized setting is a promising future direction, given the demonstrated successes of such approaches.
>
> ----
>
> ## 2. Related Work
>
> We thank the reviewer for sharing the related work discussing the intransitivity of RLHF! We have added the discussion in our revision (please see Page 1, line 45 in our updated manuscript).
>
> ----
>
> ## 3. SFT = BC = action matching is well known, but BC = action matching = trajectory distribution matching is **new**
>
> From an RL perspective, we agree with the reviewer that equating SFT to BC is not surprising.
>
> However, in RL literature, BC is typically understood as "action (marginal) distribution matching," which is known to suffer from the compounding error problem. Interestingly, we highlight that **in the context of LLM alignment, BC goes beyond action distribution matching to also include trajectory distribution matching**. This arises from the deterministic concatenation of sentences and tokens during generation (i.e., state transitions). This unique aspect allows us to unify SFT and IRL-based AfD methods within a single distribution matching framework.
>
> We believe this novel equivalence between marginal distribution matching and trajectory distribution matching is an important contribution to the community.
>
> Furthermore, this equivalence, along with the use of forward KL in establishing it, **naturally motivates our exploration of using reverse KL for distribution matching within the same framework.** This enables us to explain the differing alignment behaviors and properties induced by these distinct distribution-matching objectives.

---

> > ### Author Response · Authors · 2024-11-18
> > **Author Response to Reviewer iqxv (Part 2)**
> >
> > ## 4. Closed-form solution and insights based on it
> >
> > We thank the reviewer for highlighting this insightful point! With the closed-form expression for the BoN optimization procedure, we are equivalently optimizing for the following closed-form reward:
> >
> > $$r_{c}(y|x)=\log \bar{\pi}_{SFT}(y|x) - \log \pi_0 (y|x)$$
> >
> > Consequently, the policy optimization objective for BoN becomes
> >
> > $$\arg\max_{n} r(y_n|x) = \arg\max_{n}(\log\bar{\pi}_{SFT}(y_n|x) -\log \pi_0(y_n|x) )$$
> >
> > In such an objective, we can directly calculate the probabilities of generating any $y_n$ from those parameterized models, the parameters in the $\pi_{SFT}$ model are frozen in calculating the probabilities, and the N samples are generated by this $\pi_{SFT}$ model;
> >
> > To understand the **theoretical interpretation of such an objective**, we note
> >
> > $\max_\pi KL(\pi||\pi_0)-KL(\pi||\pi_{SFT})=\max \mathbb{E}_{y\sim\pi}[r_c(y|x)]$
> >
> > Therefore, optimizing the generation using BoN with regard to $r_c$ (i.e., the order statistics) can be interpreted as simultaneously maximizing the KL divergence between the order statistics and $\pi_0$ and minimizing the KL divergence between the order statistics and $\pi_{SFT}$. **This exactly leads to the extrapolation behavior as we desired!**
> >
> >
> > ---
> >
> > **Empirically**, the challenge lies in the computational cost of calculating those probabilities. To calculate the closed-form reward, we need 2-3 LLMs to be loaded in the memory: the LLM to be optimized $\pi$, the SFT checkpoint $\pi_{SFT}$, and the initial checkpoint $\pi_0$. Such a closed-form reward model takes 3 times more memory than using the discriminator — which can be implemented as a value head of LLMs and only has a small number of parameters to optimize.
> >
> > To empirically verify the effectiveness of such a closed-form reward function, we experiment with the Harmless dataset, results are shown in the table below:
> >
> > Table 1. Golden Reward (before normalization)
> > | Method             | N  = 2      | N = 5       | N = 10      | N = 30      | N = 50      |
> > |--------------------|-------------|-------------|-------------|-------------|-------------|
> > | **Close Form**     | 1.926 ± 0.047 | 2.191 ± 0.068 | 2.282 ± 0.065 | 2.348 ± 0.054 | 2.383 ± 0.061 |
> > | **Init - SFT**     | 1.901 ± 0.069 | 2.063 ± 0.121 | 2.171 ± 0.065 | 2.272 ± 0.101 | 2.333 ± 0.122 |
> > | **Init - Demo**    | 1.691 ± 0.106 | 1.575 ± 0.064 | 1.506 ± 0.126 | 1.362 ± 0.071 | 1.330 ± 0.058 |
> > | **SFT - Demo**     | 1.664 ± 0.111 | 1.537 ± 0.039 | 1.420 ± 0.059 | 1.330 ± 0.087 | 1.306 ± 0.084 |
> > | **Human - Pairwise** | 1.856 ± 0.104 | 1.949 ± 0.052 | 2.020 ± 0.074 | 2.059 ± 0.089 | 2.058 ± 0.060 |
> >
> > Table 2. BoN Win Rate
> > | Method             | N=2           | N=5           | N=10          | N=30          | N=50          |
> > |---------------------|---------------|---------------|---------------|---------------|---------------|
> > | **Close Form**      | 0.626 ± 0.043 | 0.739 ± 0.044 | 0.774 ± 0.028 | 0.826 ± 0.033 | 0.835 ± 0.025 |
> > | **Init - SFT**      | 0.594 ± 0.033 | 0.721 ± 0.028 | 0.752 ± 0.025 | 0.818 ± 0.021 | 0.805 ± 0.026 |
> > | **Init - Demo**     | 0.401 ± 0.029 | 0.396 ± 0.022 | 0.311 ± 0.041 | 0.231 ± 0.033 | 0.193 ± 0.023 |
> > | **SFT - Demo**      | 0.399 ± 0.030 | 0.341 ± 0.036 | 0.284 ± 0.021 | 0.225 ± 0.022 | 0.167 ± 0.014 |
> > | **Human - Pairwise**| 0.565 ± 0.033 | 0.676 ± 0.042 | 0.710 ± 0.014 | 0.735 ± 0.025 | 0.735 ± 0.018 |
> >
> >
> > We find using the **closed-form expression of the reward to perform BoN can achieve better performance than using the direct reward modeling method**, however, generating the probability takes **2 times more memory and computation as compared with the reward model parameterization method**. The closed-form solution will shine when the LLMs are small and inference with the LLMs is computationally affordable, while parameterizing the reward models will be a more efficient alternative when calculating the closed-form probability is infeasible.

---

> ### Author Response · Authors · 2024-11-18
> **Author Response to Reviewer iqxv (Part 3)**
>
> ## 5. Policy optimization
>
> We thank the reviewer’s interest in the results with parameterized policy optimization methods (e.g., PPO). In our manuscript, those results were deferred to Appendix E.7.
>
> In Figure 6, we conducted policy optimization using PPO and compared the results with those in BoN. We find that PPO outperforms BoN with a small number of N (=50) and can achieve on-par performance with BoN with N = 500.
>
> ---
>
> ## 6. Golden reward and win rate
>
> Yes, if we use the golden reward model rather than the learned ones to pick the best response out of N responses and evaluate it, it will outperform all other competitors and achieve a win rate of 1.0
>
>
> ----
>
> ## 7. The ablation study results in Figure 4
>
> In Figure 4, we experimented with different reward modeling choices to highlight the challenges analyzed in Section 3.2 — the heterogeneous problem (also known as reward hacking) in reward modeling.
>
> When the curves go down, it means using those reward modeling choices can not lead to an effective reward model — those reward models hacked the discriminative tasks and they are not good choices for reward modeling.
>
> To be more explicit, this is because those reward models focus on the incorrect aspect of responses in identifying whether they are positive samples or negative samples. Therefore, optimizing toward increasing those reward values can not lead to an increase in the golden reward values.
>
> ----
>
> ## 8. Discussion on SPIN / Direct methods
>
> Thank you for the suggestion, and we are glad to hear that our insights resonate with you. We have moved the discussion on SPIN to follow Section 3.2, as we agree that this adjustment enhances the clarity of our ideas and better distinguishes our work from related studies.
>
>
> -----
>
> **Once again, we would like to thank the reviewer for their insightful comments and thorough reading to improve our paper. Should there be any leftover concerns, please don’t hesitate to let us know and we will do our utmost to address them!**

---

> > ### Comment · Reviewer_iqxv · 2024-11-18
> > **Re:**
> >
> > Thank-you for the thorough response!
> >
> > Re: 3 -- I don't think this depends on the dynamics at all? BC is always maximizing the probability of expert trajectories. If you write out the expression for the trajectory-level KL divergence from expert to learner under stochastic dynamics, you'll observe that the dynamics cancel out in the log ratio, giving you the standard BC objective. I would argue this is a consequence of the well-known fact that KL tensorizes.
> >
> > Re: 4 -- Interesting. Intuitively, it seems like what you hope to get out of the BoN procedure is a policy that continues in the "direction" the training took the policy from the initialization (from $\pi_{init}$ to $\pi_{SFT}$). This begs the question: could we not have proceeded further along this path with more BC on the demonstration dataset? What is it about learning a discriminator / doing on policy sampling that lets one transcend offline training?
> >
> > The empirical results make sense. If you have the space, I'd suggest including the closed-form results somewhere in the paper (acknowledging the higher memory requirement). You could also frame these results as confirmation that the discriminator is trained well, as the performance is not too much worse than what you get when using the Bayes-optimal discriminator.
> >
> > Re: 5 -- Thanks, can you more explicitly mention these results in the main paper?

---

> ### Author Response · Authors · 2024-11-19
> **Thanks for your further feedback!**
>
> We thank the reviewer iqxv for their further feedback!
>
> ----
> ## On BC and Forward KL.
>
> We thank our reviewer for raising their further question on the relationship between BC and the trajectory forward KL. To better support the discussion, we would like to formally highlight the differences below:
>
> In BC, the objective is forward distribution matching of **actions**, i.e.,
>
> $$J_{BC}(\pi)=-\mathbb{E}_{(s,a)\sim \beta } \left[\log(\pi(a|s)) \right]$$
>
> In **general trajectory distributional matching**, the objective is
>
> $$J_{FKL-\tau} = KL(d^\beta(\tau|s_0)||d^\pi(\tau|s_0))= \mathbb{E}_{(\tau|s_0)\sim\beta} \left[\log d^\pi(\tau|s_0) \right]$$
>
> where we use $\tau |s_0$ to denote the trajectories starting from state $s_0$: $ \tau |s_0 =  \\{s_0, a_0, s_1, a_1,...\\} $. In general, such an objective is intractable and it is different from $J_{BC}(\pi)$ since the calculation of the trajectory density requires access to the transition probability of $p(s'|s,a), \forall s, a$.
>
> In the MDP of LLM token generation (in alignment), we know $p(s'|s,a) = 1, \mathrm{for}~ s'=\mathrm{concate}(s,a), \mathrm{and}~ 0 ~\mathrm{otherwise}$. **Therefore, with this specific context can we show the equivalence between trajectory distribution matching using forward KL and BC.** (cf. Equation (6) in our manuscript).
>
>
>
> In the IRL literature, it has been argued that the forward distribution matching on occupancy measure (joint distribution of state-action pairs) is different from the distribution matching of actions (BC) [1, 2]. Different from the IRL literature, we study trajectory matching instead since _responses_ as trajectories are more meaningful than tokens in the context of LLM alignment. Another derivation that might be interesting to our reviewer is that --- if we follow the RL literature to consider the **occupancy measure matching** (i.e., matching the distribution of **incomplete sentences**), we will have the following objective with forward KL:
>
> $$J_{FKL-(s,a)} = \left[\mathrm{KL}(\rho^\beta(s,a)||\rho^\pi(s,a)) \right] = \mathbb{E}_{(s_k,a_k)\sim\rho^\beta} [\sum^k_0
>  \log \pi(a_t|s_t)] $$
>
> $$J_{FKL-(s,a)} = \mathbb{E}_{(s_k,a_k)\sim \rho^\beta}  \left[ \frac{K-k}{K}\log \pi(a_k|s_k) \right]$$
>
> where we use $K$ to denote the maximal length of generation, and $(s_k,a_k)\sim \rho^\beta$ denotes a uniform sampling from the (incomplete) sentence-token pairs generated by the unknown demonstrator $\beta$.
>
> We can observe that in this occupancy measure matching case, using forward-KL matching also leads to a different objective from the BC loss. This weighted loss can be interpreted as follows: to align the distribution of (the uniformly sampled incomplete) sentences in the dataset, special emphasis must be placed on the correct generation of initial tokens, as they provide the foundation for generating subsequent tokens. Consequently, these initial tokens are assigned greater weights during supervised learning.
>
> ----
>
> ## Supervised Learning and Extrapolation
>
> We thank the reviewer for raising this interesting point! We would like to link such an interpretation with the iterative application of SPIN-type algorithms --- the core difference is that, in either continued supervised learning (more BC) on the demonstration dataset or in SPIN where the demonstration dataset is considered to be the positive samples, the demonstration samples strictly limit the performance of those algorithms at convergence. On the other hand, extrapolating over explicit reward models (either closed-form or through parameterization) enables further improvements.
>
> We have added those discussions and corresponding results in our updated manuscript (Added in Appendix G and referred to on page 7. Highlighted with orange text in our updated manuscript). We would like to deeply appreciate our reviewer for bringing up this insightful point!
>
> ----
>
> ## Referring to Appendix in Main Text
>
> Thank you for the suggestion on referring to additional results in the main text. In our revised manuscript, we explicitly guide our readers on page 8 (the beginning paragraph of the experiment section) for those extended results in our appendix.
>
>
>
>
> -----
> **References**
>
> [1] Ghasemipour, Seyed Kamyar Seyed, Richard Zemel, and Shixiang Gu. "A divergence minimization perspective on imitation learning methods." Conference on robot learning. PMLR, 2020.
>
> [2] Ghasemipour, Seyed Kamyar Seyed, Shane Gu, and Richard Zemel. "Understanding the relation between maximum-entropy inverse reinforcement learning and behaviour cloning." (2019).
>
> ----
> **Once again, we would like to thank the reviewer for their insightful comments and thorough reading to improve our paper. We hope the responses above have addressed the reviewer's follow-up questions. Please let us know if there are any remaining questions, and we would be happy to engage further to ensure all concerns are addressed and improve our paper!**

---

> > ### Comment · Reviewer_iqxv · 2024-11-23
> > **Re:**
> >
> > Hi,
> >
> > Apologies for the slow response on my part, crazy week.
> >
> > Re: BC vs. FKL: sorry, I think I should have been more clear in my earlier response. I was trying to note that if we consider the KL divergence between the expert and learner trajectory distributions, we can expand this as
> >
> > D_{KL}(p_{E}(\xi)||p_{\pi}(\xi))
> >
> > = \sum_{\xi \in \Xi} p_{E}(\xi) \log(p_{E}(\xi) / p_{\pi}(\xi))
> >
> > =  \sum_{\xi \in \Xi} p_{E}(\xi) \log\left(\frac{p(s_0)\prod_{h=1}^H \pi_E(a_h|s_h) T(s_{h+1}|s_h, a)}{p(s_0)\prod_{h=1}^H \pi(a_h|s_h) T(s_{h+1}|s_h, a)}\right)
> >
> > =  \sum_{\xi \in \Xi} p_{E}(\xi) \log\left(\frac{\prod_{h=1}^H \pi_E(a_h|s_h) }{\prod_{h=1}^H \pi(a_h|s_h) }\right)
> >
> > =  \sum_{\xi \in \Xi} p_{E}(\xi) [\log({\prod_{h=1}^H \pi_E(a_h|s_h) }) - \log (\prod_{h=1}^H \pi(a_h|s_h))]
> >
> > =  \sum_{\xi \in \Xi} p_{E}(\xi) [\sum_{h=1}^H \log({\pi_E(a_h|s_h) }) - \sum_{h=1}^H \log (\pi(a_h|s_h))]
> >
> > The learner has no control over the first term in this difference so we can drop it from an optimization. This leaves us with
> >
> > =  \sum_{\xi \in \Xi} p_{E}(\xi) [- \sum_{h=1}^H \log (\pi(a_h|s_h))]
> >
> > Observe that nothing inside the sum depends on the prefix before arriving at some s_h. Thus, so long as we take an expectation over a distribution that matches the marginal probability of the expert visiting s_h, we would have the same value as the above expression. This is literally what a visitation distribution / occupancy measure is. Thus:
> >
> > = H \mathbb{E}_{s, a \sim \rho_E}[- \log \pi(a_h|s_h)].
> >
> > This is the standard BC objective. So, BC is *always* minimizing forward KL. As I mentioned above, there is no need to know anything about the dynamics as they cancel out in the log ratio. We also do not need to know the dynamics to calculate the expectation as we are not actively querying the expert -- we just have sampled demos in IRL.
> >
> > Re: offline vs. online: I think I'm asking a subtler question here: for both imitation and preference-based methods, one can either use the data to learn a policy or to learn a reward model. Notably, we use the same data for both, and for LLM fine-tuning, one usually uses the same model class for both (e.g. for learning from preferences, both RMs and DPO start from an SFT checkpoint). Given we're using the same data and the same hypothesis class, there isn't a clear statistical reason RMs should generalize better than policies. And, if this RM doesn't generalize better, there is no reason to believe that an online method should do better than an offline method -- there is no "magic" in interaction. This is different than the more classic IRL setting, where one usually thinks of the reward as being learned on top of a "smaller" space of moments, which means you need fewer samples to learn well / generalize better than directly learning a policy. So, I think what I'd like to understand is what, fundamentally, about this problem makes it such that it is easier to learn a RM than a policy from essentially the same data.

---

> > > ### Author Response · Authors · 2024-11-24
> > > **Author Response to Reviewer iqxv**
> > >
> > > Thank you for the further feedback! Hope everything went smoothly :)
> > >
> > > ## On BC and FKL
> > >
> > > Thank you for the clarification! We now better understand the reviewer’s point, and we agree the reviewer is correct. From the perspective of trajectory distribution matching, the FKL minimization directly leads to the BC objective — maximizing the likelihood of a trajectory under the learner's policy is equivalent to maximizing the likelihood of each action taken in that trajectory.
> > >
> > > On the other hand, when using the conventional occupancy measure instead of the trajectory distribution matching, connecting FKL with BC needs additional information on the dynamics model. As discussed in our previous response, in the context of token generation, this leads to a _weighted version_ of the BC objective.
> > >
> > > **We have made the revision in Appendix D.2 (please refer to pages 24-25) correspondingly in our updated manuscript. We would thank the reviewer again for the inspiring discussion.**
> > >
> > > ----
> > > ## Direct methods vs. explicit reward modeling methods
> > >
> > > We thank our reviewer for further raising this point! While we do not explicitly discuss the online and offline problems in our work, we believe it would be useful to thoroughly discuss direct / two-stage methods in our related work section. In the following, we compare the direct alignment methods with explicit reward modeling methods using different perspectives:
> > >
> > >
> > > ### 1. From the Perspective of Generalization
> > >
> > > Explicit reward modeling has been shown to generalize to OOD samples better than the direct alignment methods, both theoretically and empirically [1-2]. The central insight behind this can be attributed to that learning a discriminative model can be easier than learning a generative model [3].
> > >
> > > ### 2. From the Perspective of Online and Offline
> > >
> > > The challenge of offline-ness faced by the direct alignment methods was theoretically studied in [4]. Iterative DPO methods alleviate such a problem [5] by generating on-policy responses. The core idea of using on-policy reward models in response evaluation is also studied in [6], using a data-centric off-policy evaluation perspective.
> > >
> > >
> > > ### 3. From the Perspective of Reward Hacking in RLHF
> > >
> > > There is another line of work studied using SFT objectives as regularizations for the direct alignment methods [7-9]. It is worth noting all of those works focused on preference-based alignment setups.
> > > In practice, it has been found that direct alignment methods tend to reduce probabilities in generating both preferred and dispreferred responses [10]. In direct reward modeling methods, the reward hacking problem was mainly studied from the perspective of overoptimization [11], and it has been shown that using a reward model at a 3B scale is sufficient in alleviating the problem.
> > >
> > > Besides the above differences, another unique challenge is the off-policy-ness of the expert demonstration dataset, and the potential reward hacking problem.
> > >
> > > Different from the preference-based learning methods --- where both preferred samples and dispreferred samples are generated by the LLM to be aligned (in an ideal case) --- in the AfD setup, expert demonstrations are inherently different from the LLM’s generation, therefore the heterogeneity leads to potential reward hacking problem in discriminative reward modeling. This is a new challenge and in our work, we solve it with the init-SFT reward modeling methods.
> > >
> > > ### 4. From the Perspective of Extrapolation
> > >
> > > In our work, the key difference we would like to highlight is the ability of **extrapolating over the learned reward model**: our work differs from SPIN and SFT toward the expert demonstration data because the demonstration dataset is considered to be the positive samples in those methods. As a consequence, the demonstration samples strictly *upper-bounds the performance* of those algorithms at convergence. On the other hand, extrapolating over explicit reward models (either closed-form or through parameterization) enables further improvements.
> > >
> > > **We have added those discussions in Appendix A.6 (please refer to pages 20-21) in our updated manuscript. (highlighted with orange text)**

---

> > > > ### Author Response · Authors · 2024-11-24
> > > > **Author Response to Reviewer iqxv (Cont. References)**
> > > >
> > > > **References**
> > > >
> > > > [1] Lin, Yong, et al. "On the limited generalization capability of the implicit reward model induced by direct preference optimization." arXiv preprint arXiv:2409.03650 (2024).
> > > >
> > > > [2] Xu, Shusheng, et al. "Is dpo superior to ppo for llm alignment? a comprehensive study." arXiv preprint arXiv:2404.10719 (2024).
> > > >
> > > > [3] Ouyang, Long, et al. "Training language models to follow instructions with human feedback." Advances in neural information processing systems 35 (2022): 27730-27744.
> > > >
> > > > [4] Xiong, Wei, et al. "Iterative preference learning from human feedback: Bridging theory and practice for rlhf under kl-constraint." Forty-first International Conference on Machine Learning. 2024.
> > > >
> > > > [5] Dong, Hanze, et al. "Rlhf workflow: From reward modeling to online rlhf." arXiv preprint arXiv:2405.07863 (2024).
> > > >
> > > > [6] Sun, Hao, et al. "When is off-policy evaluation useful? a data-centric perspective." arXiv preprint arXiv:2311.14110 (2023).
> > > >
> > > > [7] Liu, Zhihan, et al. "Provably mitigating overoptimization in rlhf: Your sft loss is implicitly an adversarial regularizer." arXiv preprint arXiv:2405.16436 (2024).
> > > >
> > > > [8] Gui, Lin, Cristina Gârbacea, and Victor Veitch. "BoNBoN Alignment for Large Language Models and the Sweetness of Best-of-n Sampling." arXiv preprint arXiv:2406.00832 (2024).
> > > >
> > > > [9] Pang, Richard Yuanzhe, et al. "Iterative reasoning preference optimization." arXiv preprint arXiv:2404.19733 (2024).
> > > >
> > > > [10] Pal, Arka, et al. "Smaug: Fixing failure modes of preference optimisation with dpo-positive." arXiv preprint arXiv:2402.13228 (2024).
> > > >
> > > > [11] Gao, Leo, John Schulman, and Jacob Hilton. "Scaling laws for reward model overoptimization." International Conference on Machine Learning. PMLR, 2023.
> > > >
> > > > ----
> > > > **Once again, we would sincerely thank the reviewer iqxv for their time and effort devoted to improving our paper!
> > > > We are still eager to use the remaining discussion period to fully address any additional concerns the reviewer may have.**

---

> > > > > ### Comment · Reviewer_iqxv · 2024-11-24
> > > > >
> > > > > Hi,
> > > > >
> > > > > Re: occupancy measure matching: Sorry, but matching trajectory distributions implies matching occupancy measures regardless of the dynamics. This is because an occupancy measure can be computed from a trajectory distribution (i.e. $\rho_E(s) = (1/H) \sum_{\xi \in \Xi } p_E(\xi) \sum_h^H 1[s_h = s]$). It's the other direction that isn't true: just because one matches the time-averaged state visitation frequency, this does not mean they match the path-dependent trajectory distribution.
> > > > >
> > > > > I believe the point you're trying to make is that for tree-structured problems like language where we include the entire history as state and therefore the sequence of actions we took is uniquely decodable from the state, matching occupancy measures implies matching trajectory distributions. But, BC is already matching trajectory distributions as per my previous response.
> > > > >
> > > > > Re: direct vs. reward-based methods: Yup, 1. is the answer I was looking for. I think this is the true reason your BoN procedure actually out-performs the policy it is sampling from without any new human data. It is just easier to learn a "verifier" / reward model for these tasks than a "generator" / policy. I think the other 3 points you bring up are mostly orthogonal and would suggest cutting them. Specifically, I would suggest emphasizing 1. in the paper, and mentioning this is the reason why one can reasonably believe that, without any new human data, this procedure can actually give us improved performance. I've read these references before and I don't think anyone has given a compelling explanation for why this should be the case but I think at least pointing out that this is empirically true is a good thing.

---

> > > > > > ### Author Response · Authors · 2024-11-25
> > > > > > **Author Response**
> > > > > >
> > > > > > Thank you for the further clarification!
> > > > > >
> > > > > > ---
> > > > > >
> > > > > > ### - Occupancy measure matching:
> > > > > >
> > > > > > We see the reviewer's point, and we fully agree with the reviewer on the equivalence between FKL trajectory matching and BC.
> > > > > >
> > > > > > We thank the reviewer for using the tree structure to characterize the problem class. For occupancy measure matching, we would like to use an example to clarify our point.
> > > > > > Consider the case where there is only one prompt, $\texttt{7+2}$, and the expert generates 2 tokens, '=', '9'. In this case, there are 2 actions and corresponding transitions:
> > > > > > $s_0 = \texttt{7+2}, a_0 = \texttt{=}, s_1 = \texttt{7+2=}, a_1 = \texttt{9}, s_2 = \texttt{7+2=9}$. And $\rho_E(s_1)=\rho_E(s_2)=1/2$.
> > > > > >
> > > > > > With the tree structure (deterministic dynamics), we can explicitly write $\rho_\pi(\texttt{7+2=9}) = \pi(\texttt{9}|\texttt{7+2=})\pi(\texttt{=}|\texttt{7+2})p(\texttt{7+2})$, and $\rho_\pi(\texttt{7+2=}) = \pi(\texttt{=}|\texttt{7+2})p(\texttt{7+2})$. When uniformly sampling from the states for FKL occupancy measure matching, both states get the same probability of being selected, yet $\pi(\texttt{=}|\texttt{7+2})$ is being optimized more frequently than $\pi(\texttt{9}|\texttt{7+2=})$. This leads to a different objective than trajectory distribution matching (BC) --- regardless of the dynamics --- as pointed out by the reviewer.
> > > > > >
> > > > > > In more general cases without the tree structure (or the knowledge of dynamics), [1] illustrated how the standard BC objective is different from FKL occupancy measure matching (please refer to Table 1 of [1], and Hypothesis 1 --- which was later on empirically verified in the paper.)
> > > > > >
> > > > > > ### - Discussion on Generalization
> > > > > >
> > > > > > We thank the reviewer for their further clarification and for their affirmative comments on the importance and contribution of our study. We have revised our manuscript accordingly.
> > > > > >
> > > > > > ----
> > > > > >
> > > > > > **Reference**
> > > > > >
> > > > > > [1] Ghasemipour, Seyed Kamyar Seyed, Richard Zemel, and Shixiang Gu. "A divergence minimization perspective on imitation learning methods." Conference on robot learning. PMLR, 2020.
> > > > > >
> > > > > > ---
> > > > > > Once again, we thank our reviewer for their diligent effort in reviewing and improving our paper. And please let us know if further clarifications are needed!

---

> > > > > > > ### Comment · Reviewer_iqxv · 2024-11-27
> > > > > > > **Re:**
> > > > > > >
> > > > > > > Hi,
> > > > > > >
> > > > > > > Yup, makes sense about occupancy matching and trajectory matching being different in general.
> > > > > > >
> > > > > > > I was making a separate observation: I'm fairly sure that under tree-structured dynamics, matching occupancy measures implies matching trajectory distributions. Usually, one can only say the reverse is true: that if one matches trajectory distributions, one has matched occupancy measures.
> > > > > > >
> > > > > > > I was then trying to note that given BC is minimizing trajectory-level KL, it should also, assuming a flexible enough policy class, minimize occupancy-measure KL. This is true regardless of the dynamics. So, without a compelling argument about a specific kind of mis-specification of the learner's policy class (i.e. $\pi_E \notin \Pi$), I'm not sure why one would actually care about explicitly minimizing the occupancy measure divergence, as we always get it "for free" via BC / trajectory-matching.
> > > > > > >
> > > > > > > Anyhow, I think I'm making a point a bit orthogonal to yours. I think so long as you point out that it is well-known that BC is minimizing trajectory-level KL, that this is different than optimizing occupancy-measure KL (citing [1]), but that trajectory-level matching implies occupancy matching, things should be clear enough to the readers.
> > > > > > >
> > > > > > > Thank-you for talking through this with me!

---

> > > > > > > > ### Author Response · Authors · 2024-11-29
> > > > > > > > **Thank you for reviewing our paper!**
> > > > > > > >
> > > > > > > > We appreciate the inspiring discussions with the reviewer iqxv!
> > > > > > > >
> > > > > > > > Regarding the original review, we hope the responses and discussions have addressed the outstanding questions and the reviewer would consider raising their score if those questions have been appropriately answered.
> > > > > > > >
> > > > > > > > Should there be any leftover concerns, in the time remaining, we are committed to making every effort to resolve any outstanding issues!

---

### Official Review · Reviewer_ZCzk · 2024-11-04

**Soundness:** 3
**Presentation:** 2
**Contribution:** 3
**Rating:** 6
**Confidence:** 4

**Summary:**

This paper presents Alignment from Demonstrations (AfD), a new method for aligning large language models (LLMs) using high-quality demonstration data rather than traditional preference-based reinforcement learning from human feedback (RLHF). AfD addresses issues of noisy labels, cost, and privacy by framing alignment within a Markov Decision Process, applying insights from forward and inverse reinforcement learning to create a computationally efficient, trajectory-based mechanism. The approach is validated through theoretical and empirical analyses, showing improvements on “Harmless” and “Helpful” tasks with the Anthropic HH-RLHF dataset.

**Strengths:**

1. **Innovative Use of RL Concepts:** The authors effectively integrate RL concepts—such as inverse RL, reverse KL, and forward KL divergence—into the LLM alignment framework. This combination with RLHF provides a fresh, rigorous perspective on alignment, enriching AfD’s theoretical foundation and adaptability.

2. **Reduced Dependence on Preference-Based Data:** By bypassing preference data requirements, AfD proposes a scalable alternative that minimizes interaction with human annotators while still achieving alignment, making it potentially more feasible for large-scale applications.

**Weaknesses:**

1. **Overly Complex Presentation:** The paper’s presentation is somewhat dense, with extensive theoretical detail that can make it harder to grasp the core contributions. A more streamlined focus on the main insights and practical implications of AfD could enhance clarity and accessibility for readers.

2. **Potential Overlap with Existing Methods:** The unique contribution of AfD relative to SPIN isn’t entirely clear. SPIN leverages a DPO-like objective to align LLMs directly without relying on a reward model, while AfD introduces alignment through a reward model. Clarifying the specific advantages or improvements AfD provides over methods like SPIN would strengthen the paper’s case for its distinct value.

3. **Efficiency Clarification Needed:** Although the paper suggests that AfD offers greater efficiency than traditional RLHF, it’s unclear where these efficiency gains are realized. The pseudocode presented appears similar to RLHF workflows, with steps involving reward model training and optimization. Providing more concrete details on how AfD reduces computational overhead or training time compared to RLHF would clarify the practical benefits of this approach.

**Questions:**

1. Could the authors clarify why they chose to rely primarily on the golden reward model for evaluation rather than using GPT-4 as a critic throughout in Section 4.1? Would the golden reward model alone provide a sufficiently fair or robust assessment of alignment performance, especially given GPT-4’s nuanced evaluation capabilities?

2. Could the authors clarify the key distinction between AfD and SPIN, particularly regarding their reliance on reward models? From my understanding, SPIN uses a DPO-like objective to align LLMs directly without a reward model, whereas AfD relies on a reward model for alignment. Given this, could the authors elaborate on the specific advantages AfD provides over SPIN in terms of contribution to the field?

3. The authors mention that AfD is more efficient than traditional RLHF methods, but it would be helpful to understand precisely where these efficiency gains come from. Could the authors specify which parts of the AfD process contribute to this claimed efficiency, particularly in comparison to standard RLHF?

---

> ### Author Response · Authors · 2024-11-18
> **Author Response to Reviewer ZCzk (Part 1)**
>
> We thank the reviewer for their time and thoughtful review of our paper, as well as their encouraging recognition of our innovation in providing an IRL framework for LLM alignment. Below, we would like to address their concerns and questions in turn:
>
> ----
>
> ## Q1: Evaluation with golden reward models
>
> We thank the reviewer for highlighting the need for further clarification regarding our motivation for using golden reward models in evaluation.
>
> **1. Open-Source and Reproducibility**
> On one hand, evaluating with golden reward models is significantly more cost-effective than relying on commercial APIs like GPT-4. Moreover, leveraging open-source reward models enhances reproducibility and facilitates fair comparisons in future research. This approach is a well-established practice in the reward modeling literature [1–6].
>
> **2. Evaluating with GPT-as-a-Judge**
> That said, we agree with the reviewer that incorporating GPT-as-a-judge provides valuable complementary insights into evaluating the performance of various methods. To this end, we included GPT-based evaluations in Section 4.3 (Table 2). Notably, the results obtained from GPT-based evaluation are broadly consistent with those derived from golden reward models.
>
> **3. Using both evaluation methods as cross-validation**
> Recent studies have raised concerns about the reliability of LLM-based evaluations [7–10]. To address this, we employ multiple evaluation metrics, combining golden reward models with GPT-as-a-judge. This dual approach enables a more comprehensive and reliable assessment of our proposed method.
>
> We hope this dual evaluation strategy underscores the **rigor and thoroughness of our assessment** and addresses the reviewer’s concerns.
>
> ---
>
> ## Q2: A More Direct Comparison to SPIN.
>
> We thank the reviewer for suggesting that a comparison with SPIN in the main text would be more informative than including it solely in the appendix. In the original manuscript, discussions and comparisons with SPIN were deferred to Appendix A.
>
> In the revised version, we have **moved the discussion and the illustrative example to the main text, with updates highlighted in blue.**
>
>
> **1. Achieving Super-Demonstrator Performance: A Fundamental Limitation of SPIN**
>
> A key distinction between our method and SPIN lies in the flexibility and potential of our approach to achieve **super-demonstrator performance** in LLM alignment, as empirically validated in our experiments.
>
>
> SPIN operates under the explicit assumption that the current policy (initially the SFT policy) is always weaker than the demonstrations. Consequently, at convergence, the aligned LLM's performance is upper-bounded by the performance of the demonstration dataset, as the demonstrations are consistently treated as positive examples in implicit reward modeling.
>
> In contrast, our method, rooted in Inverse RL, explicitly learns a reward model and extrapolates over it. This reward modeling mechanism allows for leveraging task scores to enhance performance beyond the demonstrators. On the other hand, naively adopting SPIN's setup—using the SFT checkpoint's generations as negative examples and the demonstrations as positive examples—can adversely impact heterogeneity and lead to suboptimal performance. As demonstrated in our experiments, the IRL approach consistently achieves **super-demonstrator performance** across both tasks.
>
>
> **2. Empirical Comparisons with SPIN**
>
> To substantiate our analytical insights, we provide empirical comparisons against SPIN in Table 4 of Appendix A.5, which we now summarize in the table below:
>
> | Task       | Demo  | SFT   | IRL (N=10) | IRL (N=30) | IRL (N=50) | SPIN (iter=1) | SPIN (iter=2) | SPIN (iter=3) |
> |------------|-------|-------|------------|------------|------------|----------------|----------------|----------------|
> | Harmless   | 1.704 | 1.785 | 2.171      | 2.272      | 2.333      | 1.769          | 1.753          | 1.747          |
> | Helpful    | 0.735 | 0.588 | 0.598      | 0.692      | 0.751      | 0.640          | 0.699          | 0.706          |
>
> The table reinforces our key insight: while SPIN focuses on imitating expert demonstration behavior, our IRL-based approach surpasses the demonstration performance, achieving significantly higher scores than the demonstrators.
>
> By moving this discussion and comparison into the main text, we aim to provide a clearer understanding of the advantages of our method over SPIN.

---

> > ### Author Response · Authors · 2024-11-18
> > **Author Response to Reviewer ZCzk (Part 2)**
> >
> > ## Q3: Clarification on the difference between RLHF and AfD.
> >
> > The reviewer is correct in pointing out that both the workflow of RLHF and AfD require an explicit reward modeling step, followed by a policy optimization step. From the computational efficiency perspective, AfD is comparable to the conventional RLHF approaches. However, we would like to highlight the superiority of AfD over the conventional RLHF approaches, which lies in the fact that AfD works with only demonstration datasets, while RLHF requires preference-based annotations.
> >
> > Please kindly let us reiterate that in practice, RLHF can suffer from the difficulty of noisy labels, high annotation costs, and privacy concerns. In contrast, AfD does not suffer from those challenges and can effectively build reward models and align LLMs.
> >
> > The 4 challenges in our introduction section elaborated those key difficulties for RLHF that can be solved by AfD, and Table 3 explains how those methods differ from the others from an RL taxonomy.
> >
> > ----
> > **References**
> >
> >
> > [1] Dong, Hanze, et al. "Raft: Reward ranked finetuning for generative foundation model alignment." arXiv preprint arXiv:2304.06767 (2023).
> >
> > [2] Gao, Leo, John Schulman, and Jacob Hilton. "Scaling laws for reward model overoptimization." International Conference on Machine Learning. PMLR, 2023.
> >
> > [3] Liu, Tianqi, et al. "RRM: Robust Reward Model Training Mitigates Reward Hacking." arXiv preprint arXiv:2409.13156 (2024).
> >
> > [4] Liu, Tianqi, et al. "Statistical rejection sampling improves preference optimization." arXiv preprint arXiv:2309.06657 (2023).
> >
> > [5] Coste, Thomas, et al. "Reward model ensembles help mitigate overoptimization." arXiv preprint arXiv:2310.02743 (2023).
> >
> > [6] Yang, Rui, et al. "Rewards-in-context: Multi-objective alignment of foundation models with dynamic preference adjustment." arXiv preprint arXiv:2402.10207 (2024).
> >
> > [7] Xu, Wenda, et al. "Pride and prejudice: LLM amplifies self-bias in self-refinement." Proceedings of the 62nd Annual Meeting of the Association for Computational Linguistics (Volume 1: Long Papers). 2024.
> >
> > [8] Chiang, Wei-Lin, et al. "Chatbot arena: An open platform for evaluating llms by human preference." arXiv preprint arXiv:2403.04132 (2024).
> >
> > [9] Dubois, Yann, et al. "Length-controlled alpacaeval: A simple way to debias automatic evaluators." arXiv preprint arXiv:2404.04475 (2024).
> >
> > [10] Zheng, Xiaosen, et al. "Cheating automatic llm benchmarks: Null models achieve high win rates." arXiv preprint arXiv:2410.07137 (2024).
> >
> > -----
> >
> > **Once again, we thank our reviewer for their insightful comments in improving our paper. Should there be any leftover concerns, please let us know and we will do our utmost to address them!**

---

> ### Author Response · Authors · 2024-11-29
> **Further Feedback Welcome!**
>
> Once again, we would thank reviewer ZCzk for their time and effort devoted to reviewing and improving our paper.
>
> In our author response **posted 11 days ago**, we have addressed each of our reviewers' concerns in a detailed, point-by-point manner. We hope the responses have addressed the outstanding questions and the reviewer would consider raising their score if those questions have been appropriately answered.
>
> As the author-reviewer discussion phase nears its conclusion, we would like to kindly ask if there are any remaining concerns or additional feedback that we can address. In the limited time remaining, we are committed to making every effort to resolve any outstanding issues!

---

> ### Author Response · Authors · 2024-12-02
> **Dear Reviewer ZCzk,**
>
> We thank the reviewer for their time and thoughtful review of our paper. As the author-reviewer discussion window is narrowing to its end, we would like to summarize the reviewer’s main concerns, and how our previous responses addressed those concerns.
>
> ----
>
> ### 1. Motivations of golden reward model evaluation
>
> In their original review, our reviewer ZCzk pointed out that it would be helpful to include further clarification regarding the motivation for using golden reward models for evaluation.
>
> In our response, we highlighted the motivations of **Open-Source and Reproducibility**; and **Comprehensive evaluation methods using both the golden reward model and GPT-as-a-critic**.
>
> ---
> ### 2. A more direct comparison to SPIN.
>
> In their original review, reviewer ZCzk recommended including a more direct comparison to SPIN.
>
> In our response, we
> - 1. Discussed and compared our method with SPIN in Appendix A.
> - 2. Moved the discussion and the illustrative example to the main text in our revised manuscript
> - 3. Highlighted the key difference between our method and SPIN is the potential of _Achieving Super-Demonstrator Performance_
> - 4. Empirically, we provided the following empirical results in Table 4 of Appendix A.5.
>
> | Task       | Demo  | SFT   | IRL (N=10) | IRL (N=30) | IRL (N=50) | SPIN (iter=1) | SPIN (iter=2) | SPIN (iter=3) |
> |------------|-------|-------|------------|------------|------------|----------------|----------------|----------------|
> | Harmless   | 1.704 | 1.785 | 2.171      | 2.272      | 2.333      | 1.769          | 1.753          | 1.747          |
> | Helpful    | 0.735 | 0.588 | 0.598      | 0.692      | 0.751      | 0.640          | 0.699          | 0.706          |
>
> ---
> ### 3. Difference between RLHF and AfD.
>
> In their original review, reviewer ZCzk asked questions centering on the difference between RLHF and AfD.
>
> In our response, we highlight that AfD and RLHF differ in the dataset they work with — AfD works with **expert demonstration dataset**, while RLHF works with **pairwise preference dataset**.
> While RLHF can suffer from the difficulty of noisy labels, high annotation costs, and privacy concerns, AfD does not suffer from those challenges and can effectively build reward models and align LLMs.
>
>
> -----
>
>
> Once again, we thank our reviewer for their insightful comments in improving our paper.
> In the limited author-reviewer discussion period, we are still eager to address any further concerns from our reviewer.

---

> > ### Comment · Reviewer_ZCzk · 2024-12-02
> >
> > After careful consideration of the authors’ responses and the additional experiments provided, I have decided to adjust my rating to a 6. The responses have adequately addressed some of my initial concerns regarding the clarity and novelty of the contributions.
> >
> > While the concepts presented in the paper build upon existing ideas, the methods for explicit reward modeling and achieving super-demonstrator performance have been distinguished from prior work. The additional experiments across various datasets have helped to solidify the empirical foundation of the method.
> >
> > However, while the paper contributes meaningfully to the field, I believe there remains potential for deeper exploration and clearer articulation of the novel aspects. Thus, my recommendation reflects a positive yet cautious endorsement of the paper’s acceptance.
> >
> > **Suggestions for Improvement:**
> > - Clarify and emphasize the novel contributions more distinctly in the main text to better differentiate this work from existing literature.
> > - Consider condensing the background sections to allow more space for detailing the methodology and experimental insights, which are crucial for demonstrating the unique value of the proposed approach.
> >
> > **Suggestions for Improvement:**
> > - Focus more on the core contributions in the main text to highlight the novelty of the method.
> > - Reduce background content if necessary to provide more space for detailed discussion of the proposed approach and its implications.

---

### Official Review · Reviewer_5fua · 2024-11-05

**Soundness:** 3
**Presentation:** 2
**Contribution:** 2
**Rating:** 5
**Confidence:** 4

**Summary:**

This work studies the large language model (LLM) alignment problem in the learning from demonstration framework. Under this framework, the widely adopted supervised finetuning (SFT) paradigm can be interpreted as matching the policy distribution and an unknown expert demonstration distribution with the forward KL distance measure. The authors then propose to consider distribution matching with the reverse KL distance. This problem has been studied in the imitation learning literature. A standard method is to train a discriminator to distinguish the policy trajectories and the expert trajectories and then train the policy by reinforcement learning with reward signals derived from the discriminator's outputs. This work adopts this method in the context of LLM alignment and evaluates it empirically on the Harmless and Helpful dataset. Experiment results show that the proposed method performs better than or on par with the SFT baseline and learning from human annotated preference data.

**Strengths:**

This paper does a good job at interpreting LLM alignment under the learning from demonstration framework. It successfully frames the alignment problem in the language of distribution matching. The idea of using the reverse KL distance follows naturally.

The authors identify the heterogeneity problem in the naive adoption of the discriminator-as-reward method and propose the Init-SFT RM as a solution to mitigate the heterogeneity gap. Init-SFT RM demonstrates strong performance in the experiments. This idea provides insights into learning from demonstration and can be applied to a broader class of problems beyond LLM alignment.

**Weaknesses:**

Important details of the method is missing from the main text. Section 3.2 talks about extrapolating the learned learned reward models but does not provide any detail on how it works in the context of alignment. Perhaps as a consequence, the results presented in Section 4.3 are confusing to me. It looks like the only difference to Section 4.2 is the evaluation metric being GPT4 as a judge rather than the golden reward model.

Another weakness of this work is the lack of understanding of the behavior of the proposed method. The distinction between forward KL distance and reverse KL distance lead to two different methods in SFT and discriminator-as-reward. The authors also discussed the mass-covering and mode-seeking behavior in Section 3. One natural question to ask here is how it impacts the behavior of the alignment algorithms and if they yield different outcomes. However, the discussion in Section 4 is rather hand-wavy. The authors simply characterize the harmless dataset and the helpful dataset as less divergent and more divergent. I think a deeper analysis on the mass-covering and mode-seeking behavior in alignment can greatly improve this work.

In terms of writing, citation format is misused through the manuscript. Please proofread the paper carefully and use the proper citation command (e.g., \citep{} vs \citet{}) in the revision.

**Questions:**

1. Could the authors clarify on why this work falls into the inverse reinforcement learning category? I might be wrong, but my understanding of inverse reinforcement learning is about uncovering the underlying reward function from expert trajectories. This work is not about finding the hidden "true reward function" but about matching the demonstration distribution. Thus I am confused by the use of the term "inverse reinforcement learning".

2. In a typical alignment pipeline, learning from annotated preference data like RLHF comes after SFT. RLHF often yields mode-seeking distributions in contrast to SFT. Could the authors comment on the compatibility of the proposed method and RLHF? Should we expect RLHF to provide further improvement given that the AfD method is already mode-seeking?

---

> ### Author Response · Authors · 2024-11-18
> **Author Response to Reviewer 5fua (Part 1)**
>
> We thank the reviewer for their time and thoughtful review of our paper, as well as their encouraging recognition of our contribution to streamlining LLM alignment tasks through a distributional matching perspective. Below, we address the concerns and questions raised in turn:
>
> ----
> ## 1. Details for building reward models
>
> In our implementation, we use Equation (11) to train a discriminator, where the logits of the discriminator correspond to the reward values as defined in Equation (12). Specifically:
>
>
> $D_\phi(y|x) = \sigma(\texttt{logits}(y|x)) :=\sigma( r(y|x))$
>
> here, $D_\phi(y|x)$ is a binary classifier trained to distinguish samples generated by the SFT model from those produced by the initial model.
>
> To aid readers in understanding our implementation, we provide Algorithm 1 in Appendix E.1.
>
> Please let us know if there are further specific questions about any aspect of the reward modeling implementation.
>
> ---
>
> ## 2. Objective of Section 4.3
>
> We appreciate the reviewer for highlighting the need for clarification regarding the objective of Section 4.3.
>
> The primary goal of Section 4.3 is to evaluate our proposed method using GPT-4 as a judge—a widely adopted and complementary evaluation metric. In our work, we employ a **dual evaluation** approach, utilizing both golden reward models and GPT-4 as evaluative benchmarks. The rationale for this dual approach is as follows:
>
> - **Golden Reward Models**: Evaluating with golden reward models is significantly more cost-effective than relying on commercial APIs like GPT-4. Moreover, using open-source reward models enhances the reproducibility of our work and enables fair comparisons in future research. This practice aligns with established standards in the reward modeling literature [1–6].
> - **GPT-4 as a Judge**: Incorporating GPT-4-based evaluation provides additional insights into the performance of various methods, complementing golden reward evaluations. As shown in Section 4.3 (Table 2), our results indicate that GPT-4-based evaluations are broadly consistent with those derived from golden reward models.
>
> To address this point, we have revised the description in Section 4.3 to improve clarity and ensure our evaluation approach is well-articulated.
>
> ----
>
> ## 3. Why Our Work Falls into the IRL Method Category.
>
> We thank the reviewer for raising the question about method categorization.
> In the RL literature, both Imitation Learning (IL) and Inverse RL (IRL) aim to learn a policy from a demonstration dataset, typically involving trajectory distribution matching [e.g., 7,8]. The key distinction is that **IRL explicitly learns a reward model, while IL does not.**
> - In IL, the objective is to match the behavior of the learned policy with the demonstrator.
> - In IRL, an intermediate step involves building a reward model from the demonstration dataset, which is then used to optimize a policy to match—or even surpass—the demonstrator's performance.
>
> Due to space constraints, we deferred the detailed discussion of these differences to Table 3 and Appendix A in the manuscript.
>
> In our work, we **explicitly build a reward model** using the demonstration dataset, hence the method falls to the category of IRL rather than IL. As has been demonstrated in our experiments, such an explicit reward modeling step is essential for **achieving super-demonstrator performance**, rather than merely matching their performance. (cf. Figure 4.)

---

> > ### Author Response · Authors · 2024-11-18
> > **Author Response to Reviewer 5fua (Part 2)**
> >
> > ## 4. In-depth discussions on the forward and reverse KL
> >
> > We appreciate the reviewer’s interest in further discussion about forward and reverse KL divergences. Below, we provide additional details:
> >
> > **1. Without Our IRL-Based Method, SFT is the Only Option.**
> > When alignment relies solely on a demonstration dataset, the standard approach is SFT, which corresponds to forward-KL-based distribution matching. SFT inherently exhibits mass-covering behavior. In contrast, our proposed AfD algorithm, derived from reverse-KL distribution matching, enables mode-seeking behavior. This provides a novel alternative for using demonstration data in alignment.
> >
> > **2. Extrapolating Over Reward Models.**
> > Our work emphasizes that forward-KL-based SFT can effectively match expert performance depending on the datasets and tasks, as demonstrated in Section 4.1. However, reverse-KL-based methods, which rely on explicit reward modeling, **consistently improve performance beyond the baseline set by SFT**. This distinction isolates the gains attributable to reward modeling.
> >
> > **3. Reward Modeling Enables Inference-Time Optimization.**
> > From a practical perspective, reverse-KL-based reward modeling supports inference-time optimization, whereas forward-KL-based supervised learning objectives do not enable further performance improvements post-training.
> >
> > **4. Empirical Demonstration of Super-Demonstrator Performance.**
> > Correspondingly, our experiments using different divergence measures (i.e., the forward and reverse KL) aim at 1. highlighting properties of those different objectives (mode seeking and mass covering), and 2. Empirically demonstrate that explicit reward modeling and IRL can achieve super-demonstrator performance. Our experiments in Sections 4.1 and 4.2 illustrate two key findings:
> > - SFT (forward-KL matching) can sometimes achieve super-demonstrator performance.
> > - Reward modeling (reverse-KL matching) is consistently useful for achieving further gains.
> >
> > ----
> >
> > ## 5. Compare Workflows in AfD and RLHF
> >
> > We thank the reviewer for their insightful questions about the workflows in AfD and RLHF.
> >
> > **1. AfD is an alternative to RLHF**
> > AfD serves as a practical **alternative to RLHF** when preference annotations are unavailable due to high costs, noisy labeling, or data-sharing restrictions (e.g., privacy concerns). For instance, in healthcare applications, it is often infeasible to collect preference annotations for medical prescriptions, but expert demonstration data may be available.
> >
> >
> > **2. Summarize the differences in a table.**
> > To enhance clarity, we summarize the differences between AfD and RLHF workflows in the table below:
> >
> >
> > | Alignment Methods | SFT Data | IRL Data  | Workflow|
> > |---|---|---|----|
> > | AfD| Demonstration| Demonstration | SFT + RM + PO  |
> > | RLHF | Positive Sample  | Preference Pairs | SFT + RM + PO  |
> >
> > Both methods share SFT, reward modeling (RM), and policy optimization (PO) stages. While RLHF relies on preference annotations, **AfD operates solely with demonstration data**. Notably, both approaches face the challenge of off-policy data: In RLHF, this was solved through online iterative annotation [9], while in our AfD with expert demonstrations, we introduce the init-SFT reward modeling method to address such a challenge effectively.
> >
> > ---
> >
> > ## 6. Citation Format
> >
> > We thank the reviewer for pointing out the citation commands. We have updated the formats accordingly.
> >
> > ---
> >
> > **References**
> >
> > [1] Dong, Hanze, et al. "Raft: Reward ranked finetuning for generative foundation model alignment." arXiv preprint arXiv:2304.06767 (2023).
> >
> > [2] Gao, Leo, John Schulman, and Jacob Hilton. "Scaling laws for reward model overoptimization." International Conference on Machine Learning. PMLR, 2023.
> >
> > [3] Liu, Tianqi, et al. "RRM: Robust Reward Model Training Mitigates Reward Hacking." arXiv preprint arXiv:2409.13156 (2024).
> >
> > [4] Liu, Tianqi, et al. "Statistical rejection sampling improves preference optimization." arXiv preprint arXiv:2309.06657 (2023).
> >
> > [5] Coste, Thomas, et al. "Reward model ensembles help mitigate overoptimization." arXiv preprint arXiv:2310.02743 (2023).
> >
> > [6] Yang, Rui, et al. "Rewards-in-context: Multi-objective alignment of foundation models with dynamic preference adjustment." arXiv preprint arXiv:2402.10207 (2024).
> >
> > [7] Ho, Jonathan, and Stefano Ermon. "Generative adversarial imitation learning." Advances in neural information processing systems 29 (2016).
> >
> > [8] Fu, Justin, Katie Luo, and Sergey Levine. "Learning robust rewards with adversarial inverse reinforcement learning." arXiv preprint arXiv:1710.11248 (2017).
> >
> > [9] Xiong, Wei, et al. "Iterative preference learning from human feedback: Bridging theory and practice for rlhf under kl-constraint." Forty-first International Conference on Machine Learning. 2024.
> >
> >
> > -----
> >
> > **Once again, we thank our reviewer for their insightful comments in improving our paper. Should there be any leftover concerns, please let us know and we will do our utmost to address them!**

---

> > > ### Author Response · Authors · 2024-11-29
> > > **Dear Reviewer 5fua**
> > >
> > > Once again, we would thank reviewer 5fua for their time and effort devoted to reviewing and improving our paper.
> > >
> > > In our author response **posted 11 days ago**, we have addressed each of our reviewers' concerns in a detailed, point-by-point manner. We hope the responses have addressed the outstanding questions and the reviewer would consider raising their score if those questions have been appropriately answered.
> > >
> > > As the author-reviewer discussion phase nears its conclusion, we would like to kindly ask if there are any remaining concerns or additional feedback that we can address. In the limited time remaining, we are committed to making every effort to resolve any outstanding issues!

---

> ### Author Response · Authors · 2024-12-02
> **Dear Reviewer 5fua**
>
> We thank the reviewer for their time and thoughtful review of our paper. As the author-reviewer discussion window is narrowing to its end, we would like to summarize the reviewer’s main concerns, and how our previous responses addressed those concerns.
>
> ---
>
> ### 1. Building reward models
> In the original review, reviewer 5fua asked about the details of reward modeling.
>
> In our response, the following explanations have been provided:
> - 1. Detailed the implementation using Equation (11);
> - 2. Explained why the logits of classifiers could be used as reward signals;
> - 3. Provided Algorithm 1 in Appendix E.1;
>
> ### 2. Section 4.3
>
> In the original review, reviewer 5fua raised their concern on the objective of section 4.3 (experiment section where we use GPT4-as-critic for reward model evaluation)
>
> In our response, we highlighted the objective of leveraging a **dual evaluation** approach in our work is to enhance the reliability our conclusions.
>
> In addition, as per suggested by Reviewer wScp, we additionally experimented using the GPT3.5 demonstrations (using golden reward models for evaluation) and UltraFeedback (using GPT4o for evaluation). All of our experimental designs aim at enhancing the reliability of the conclusions in our work.
>
> ### 3. Link to IRL literature
>
> In the original review, reviewer 5fua raised the question of _Why this paper falls into the IRL method category._
>
> In our response, we highlighted the usage of an explicit reward modeling step in our method making our method an IRL algorithm. To further enhance the clarity, we discussed the key distinctions between Inverse RL and Imitation Learning. We would refer to Table 3, Figure 4, and Appendix A for more details in our updated manuscript!
>
>
> ### 4. On the forward and reverse KL
>
> In the original review, reviewer 5fua requested an additional discussion on _the forward and reverse KL_.
>
> In our response, we discussed the key insight of using the forward and reverse KL in our distributional matching objective from the following 4 perspectives:
>
> - 1. Leveraging SFT-format data beyond SFT.
>
> - 2. Mode-seeking behavior with learned reward models.
>
> - 3. Enabling inference-time optimization with reward models.
>
> - 4. Achieving super-demonstrator performance.
>
>
> ### 5. Workflow in AfD and RLHF
>
> In the original review, reviewer 5fua raised a question on comparing _AfD and RLHF_.
>
> In our response, we answered this question first by highlighting that **AfD is an alternative to RLHF** rather than an intermediate step. We then provided a comparison table that summarizes the key difference between the workflows of AfD and RLHF: Both methods have Reward Modeling and Policy Optimization stages, AfD works on **demonstration data**, and RLHF works on **pair-wise preference data**.
>
> ### 6. Citation format
>
> We thank reviewer 5fua for pointing out the citation commands. In our updated manuscript, we have updated the reference format.
>
>
> -----
>
> Once again, we thank our reviewer for their insightful comments in improving our paper.
> In the limited author-reviewer discussion period, we are still eager to address any further concerns from our reviewer.

---

### Author Response · Authors · 2024-11-22
**Futher Feedback Welcome!**

We sincerely thank our reviewers for their time and effort in reviewing our paper and for providing thoughtful feedback to help improve our work.

We are particularly grateful for their recognition of our contributions to alignment research (Reviewer 5fua, ZCzk), as well as their appreciation of the unified IRL framework we proposed for alignment (Reviewer wScp, ZCzk). We also would like to thank Reviewer iqxv for the encouraging comments on our presentation and writing, and Reviewer 5fua for their affirmative comments on our empirical results.

In our author response, we have addressed each of our reviewers' concerns in a detailed, point-by-point manner. We hope the responses have addressed the outstanding questions and the reviewer would consider raising their score if those questions have been appropriately answered. As the author-reviewer discussion phase nears its conclusion, we would like to kindly ask if there are any remaining concerns or additional feedback that we can address. In the limited time remaining, we are committed to making every effort to resolve any outstanding issues!


Best regards,

Authors

---

### Meta-Review · Area_Chair_1C3y · 2024-12-22

**Metareview:**

This paper studies LLM alignment from demonstrations. They used ideas from inverse RL, proposed the AfD (Alignment from Demonstrations) approach, and showed promising performance in empirical evaluation.

Strengths:
This paper brings different concepts together into the alignment problem and provides interesting viewpoints.

Weaknesses:
The biggest issue with this paper is about its novelty. Using inverse RL in the alignment problem has been studied for a while, and some reviewers are concerned about what new contribution this paper brings to the community.

The reviewers remained unconvinced during the rebuttal period. I agree with their concerns about novelty, which isn't well addressed in the current version. Therefore, this paper isn't ready for publication at ICLR in its present form.

**Additional Comments On Reviewer Discussion:**

The authors and reviewers seem to have significant disagreement about the assessment, especially regarding the novelty and contribution of this paper when compared with existing work. There has been extensive discussion during the rebuttal period. However, the disagreement and concerns still remain.

---

### Decision · Program_Chairs · 2025-01-22

Reject